# Overcoming Non-stationary Dynamics with Evidential Proximal Policy Optimization

**Abdullah Akgül**                                                    *akgul@imada.sdu.dk*
*Department of Mathematics and Computer Science*
*University of Southern Denmark*

**Gulcin Baykal**                                                     *baykalg@imada.sdu.dk*
*Department of Mathematics and Computer Science*
*University of Southern Denmark*

**Manuel Haußmann**                                                 *haussmann@imada.sdu.dk*
*Department of Mathematics and Computer Science*
*University of Southern Denmark*

**Melih Kandemir**                                                  *kandemir@imada.sdu.dk*
*Department of Mathematics and Computer Science*
*University of Southern Denmark*

**Reviewed on OpenReview:** *https://openreview.net/forum?id=KTfTwxsVNE*

## Abstract

Continuous control of non-stationary environments is a major challenge for deep reinforcement learning algorithms. The time-dependency of the state transition dynamics aggravates the notorious stability problems of model-free deep actor-critic architectures. We posit that two properties will play a key role in overcoming non-stationarity in transition dynamics: (i) preserving the plasticity of the critic network and (ii) directed exploration for rapid adaptation to changing dynamics. We show that performing on-policy reinforcement learning with an evidential critic provides both. The evidential design ensures a fast and accurate approximation of the uncertainty around the state value, which maintains the plasticity of the critic network by detecting the distributional shifts caused by changes in dynamics. The probabilistic critic also makes the actor training objective a random variable, enabling the use of directed exploration approaches as a by-product. We name the resulting algorithm *Evidential Proximal Policy Optimization (EPPO)* due to the integral role of evidential uncertainty quantification in both policy evaluation and policy improvement stages. Through experiments on non-stationary continuous control tasks, where the environment dynamics change at regular intervals, we demonstrate that our algorithm outperforms state-of-the-art on-policy reinforcement learning variants in both task-specific and overall return.

## 1 Introduction

Most deep reinforcement learning algorithms are developed assuming stationary transition dynamics, even though in many real-world applications the transition distributions are time-dependent, i.e., *non-stationary* (Thrun, 1998). The non-stationarity of state transitions makes it essential for the agent to keep updating its policy. For example, a robotic arm may experience wear and tear, leading to changes in the ability of its joints to apply torque, or an autonomous robot navigating a terrain with varying ground conditions, such as friction, inclination, and roughness. In such environments, an agent can maintain high performance only by continually adapting its policy to changes. On-policy algorithms, such as Proximal Policy Optimization (PPO) (Schulman et al., 2017), are particularly well-suited for non-stationary environments because they rely

solely on data from the most recent policy, ensuring policy improvement through sufficiently small updates (Kakade & Langford, 2002). This makes PPO an attractive choice for applications ranging from physical robotics (Melo & Máximo, 2019) to fine-tuning large language models (Touvron et al., 2023; Achiam et al., 2023; Zheng et al., 2023). Agents designed for open-world, non-stationary environments have to continually learn throughout their entire lifecycle, not just during a fixed training phase. Time-dependent changes in state transition dynamics result in non-stationary Markov decision processes (MDPs), where existing reinforcement learning algorithms often struggle to adapt effectively.

We posit that the simultaneous presence of two key features is essential for overcoming the challenges caused by non-stationarity in deep reinforcement learning:

*(i)* **Maintaining the plasticity of the critic network:** Plasticity refers to the ability of a neural network to change its wiring in response to new observations throughout the complete learning period. Deep reinforcement learning algorithms have been reported to suffer from the loss of plasticity in non-stationary settings by a vast body of earlier work (Dohare et al., 2021; Lyle et al., 2022; Nikishin et al., 2022; Abbas et al., 2023; Dohare et al., 2023; Lyle et al., 2023; Dohare et al., 2024; Lee et al., 2024; Moalla et al., 2024; Chung et al., 2024; Kumar et al., 2025; Lyle et al., 2025).

*(ii)* **Ensuring directed exploration for rapid adaptation to changing dynamics:** Directed exploration determines the degree of exploration based on an estimated uncertainty of an unobserved state. In this way, the agent prioritizes underexplored, hence more informative, areas of the state-action space, thereby improving its sample efficiency. Directed exploration is instrumental in fast-changing non-stationary environments where the agent has limited time to adapt to each new condition (Kaufmann et al., 2012; Besbes et al., 2014; Zhao et al., 2020).

We hypothesize that both sustained plasticity and directed exploration can be achieved by quantifying the uncertainty around the value function. An agent equipped with a probabilistic value function will systematically reduce the uncertainty of its value predictions as it collects more data. When confronted with a change in environment dynamics, the value function output will make predictions with reduced confidence. The increased uncertainty will increase the critic training loss, thereby keeping the training process active. Furthermore, the probabilistic value predictor will make it possible to assign uncertainty estimates to the policy training objective, which can in turn be used as an exploration bonus to direct the policy search toward underexplored areas of the state-action space.

> **Our Hypothesis:** *Equipping an agent with a mechanism to quantify the uncertainty of the value function enables it to (i) preserve plasticity and (ii) explore effectively under non-stationary dynamics.*

Guided by the above hypothesis, we adopt *Evidential Deep Learning* (Sensoy et al., 2018) as a well-suited framework for learning probabilistic value functions. Evidential deep learning suggests modeling the uncertainty of each data point by a Bayesian data-generating process where the hyperparameters of the prior distribution are determined by input-dependent functions. The likelihood and priors are chosen as conjugate pairs to keep the calculation of the data point-specific posterior and the marginal likelihood analytically tractable. The prior hyperparameter functions are modeled as deep neural networks, the parameters of which are learned by empirical Bayes. Evidential approaches are observed to deliver high-quality uncertainty estimates in both regression (Amini et al., 2020) and classification (Kandemir et al., 2022) settings.

Figure 1 illustrates the learning profiles of on-policy deep actor-critics in a continuous control task with non-stationary dynamics. Plain PPO, its recent extension to non-stationary environments (Moalla et al., 2024), and a state-of-the-art variant equipped with directed exploration (Yang et al., 2024b) all lose their adaptation capability at early stages of training. Conversely, our evidential version and its extension to directed exploration quickly adapt to new tasks. We posit that our new method, called *Evidential Proximal Policy Optimization (EPPO)*, brings such a performance boost as it fulfills both requirements of our hypothesis above. Our contributions are as follows:

*(i)* We apply evidential deep learning for the first time to uncertainty-aware modeling of the value function in an on-policy deep actor-critic architecture. Our solution prescribes a hierarchical Bayesian generative process that maps state observations to hyperpriors.

*(ii)* We use evidential value learning to develop two methods for constructing a probabilistic extension of the *generalized advantage estimator* (Schulman et al., 2016). We demonstrated that performing directed exploration based on the probabilistic advantage estimators brings a consistent performance improvement.

*(iii)* Due to the absence of a widely adopted benchmark, we introduce two new experimental designs tailored to evaluate the adaptation capabilities of continuous control agents to rapidly changing environment conditions. We benchmark our approach against two state-of-the-art PPO variants and observe that it outperforms them in the majority of cases.

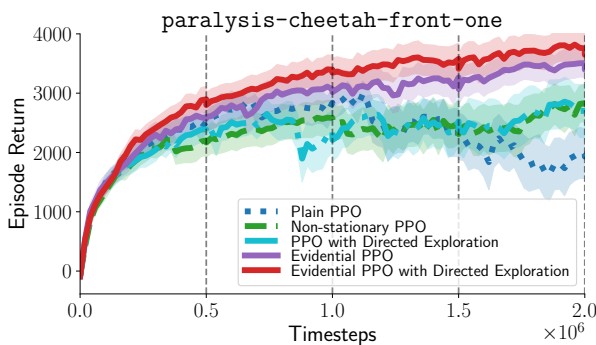

Figure 1: PPO, its non-stationary extension, and PPO equipped with directed exploration all lose their adaptation capability after 1 million steps. In contrast, evidential PPO variants continue to improve, and directed exploration further enhances evidential PPO's performance. See Section B.4 for details.

## 2 Background

### 2.1 On-policy deep actor-critics

We define an infinite-horizon MDP as a tuple $\mathcal{M} \triangleq \langle \mathcal{S}, \mathcal{A}, P, r, \rho_0, \gamma \rangle$, where $\mathcal{S}$ represents the state space and $\mathcal{A}$ denotes the action space. Let $P$ be the state transition probability distribution such that $s' \sim P(\cdot|s,a)$ where $s \in \mathcal{S}$ and $a \in \mathcal{A}$. We assume a deterministic, time-homogeneous reward function $r : \mathcal{S} \times \mathcal{A} \to \mathbb{R}$ to facilitate presentation but without loss of generality. We denote the initial state distribution as $s_0 \sim \rho_0(\cdot)$ and the discount factor as $\gamma \in (0,1)$. We consider non-stationary environments with time-dependent state transition probabilities, i.e., $P_t(\cdot|s,a)$ for a time index $t$, where the non-stationarity arises from these changes. We consider stationary stochastic policies defined as $a \sim \pi(\cdot|s)$. We use the following standard definitions of the action-value function $Q^\pi$, the value function $V^\pi$, and the advantage function $A^\pi$:

$$Q^\pi(s_t, a_t) \triangleq \mathbb{E}_{\substack{s_{t+1:\infty}, \\ a_{t+1:\infty}}} \left[ \sum_{l=0}^{\infty} \gamma^l r_{t+l} \right], \quad V^\pi(s_t) \triangleq \mathbb{E}_{\substack{s_{t+1:\infty}, \\ a_{t:\infty}}} \left[ \sum_{l=0}^{\infty} \gamma^l r_{t+l} \right], \quad A^\pi(s_t, a_t) \triangleq Q^\pi(s_t, a_t) - V^\pi(s_t),$$

where expectations are taken over trajectories induced by the policy $\pi$ and $r_{t+l} \triangleq r(s_{t+l}, a_{t+l})$. The colon notation $a : b$ refers to the inclusive range $(a, a+1, \ldots, b)$. We denote by $G_t \triangleq \sum_{l=0}^{\infty} \gamma^l r_{t+l}$ the discounted sum of rewards.

We focus our study on on-policy deep actor-critic algorithms. We adopt PPO (Schulman et al., 2017) as the state-of-the-art representative of the conservative policy iteration approaches (Kakade & Langford, 2002). This algorithm family has been adopted in real-world scenarios due to its relative robustness stemming from the conservative policy updates that promote slower but more stable training. Prime examples include the control of physical robotic platforms (Lopes et al., 2018; Melo & Máximo, 2019) and fine-tuning large language models (Christiano et al., 2017; Bai et al., 2022; Touvron et al., 2023; Achiam et al., 2023; Zheng et al., 2023). PPO is a policy gradient method that updates the policy using a surrogate objective, ensuring that policy updates remain constrained to ensure an average policy improvement (Schulman et al., 2015). We follow the established practice and adopt the clipped objective as the surrogate function. PPO updates its policy $\pi_\theta$, parametrized by $\theta \in \Theta$:

$$\mathcal{L}_{\text{clip}}(\theta) = \mathbb{E}_{(s,a) \sim \pi_{\text{old}}} \left[ \min \left( \frac{\pi_\theta(a|s)}{\pi_{\text{old}}(a|s)} \hat{A}^{\pi_{\text{old}}}(s,a), \text{clip} \left( \frac{\pi_\theta(a|s)}{\pi_{\text{old}}(a|s)}, 1-\epsilon, 1+\epsilon \right) \hat{A}^{\pi_{\text{old}}}(s,a) \right) \right],$$

where $\hat{A}^{\pi_{\text{old}}}(s,a)$ is an estimate of the advantage function, and $\text{clip}(\frac{\pi_\theta(a|s)}{\pi_{\text{old}}(a|s)}, 1-\epsilon, 1+\epsilon)$ bounds the probability ratio within the range $[1-\epsilon, 1+\epsilon]$ for $\epsilon > 0$. PPO approximates the value function with $V_\phi$ parametrized by

$\phi \in \Phi$. It uses the squared-error loss $\mathcal{L}_{\text{VF}}(\phi) = \mathbb{E}_{s_t}\left[(V_\phi(s_t) - G_t)^2\right]$ to learn $V_\phi$. The learned $V_\phi$ is then used to compute advantage estimates, guiding policy updates for more stable and efficient learning.

Modern PPO implementations use *Generalized Advantage Estimation (GAE)* (Schulman et al., 2016), which is a technique for computing advantage estimates. This method helps reduce the variance in the return estimate while enabling step-wise updates via bootstrapping. GAE constructs the advantage function using a weighted sum of multi-step temporal-difference errors. Let the temporal-difference residual at time step $t$ be $\delta_t \triangleq r_t + \gamma V_\phi(s_{t+1}) - V_\phi(s_t)$. The GAE estimate is defined as the exponentially weighted sum of temporal difference residuals:

$$\hat{A}_t^{\text{GAE}(\lambda),\pi} = \sum_{l=0}^{\infty}(\gamma\lambda)^l \delta_{t+l}, \tag{1}$$

where $\lambda \in [0,1]$ is a hyperparameter that controls the bias-variance trade-off. GAE provides a flexible mechanism for estimating advantages, allowing reinforcement learning algorithms to achieve improved stability and faster convergence (Schulman et al., 2015; 2017).

**Loss of plasticity mitigation.** Many deep reinforcement learning algorithms suffer from loss of plasticity under non-stationarity, attributed to several factors. Dohare et al. (2021; 2023; 2024) treat the loss of plasticity in PPO primarily as an optimization challenge. To address this, Dohare et al. (2021; 2024) propose a continual backpropagation method to counteract decaying plasticity, while Dohare et al. (2023) introduce a non-stationary adaptation of the Adam optimizer (Kingma & Ba, 2015), balancing exponential decay rates for moment estimates (Lyle et al., 2023). Another line of work employs regularization to preserve plasticity, such as constraining learned features to remain close to their initial values (Lyle et al., 2022), penalizing parameter drift relative to initializations (Kumar et al., 2025), promoting weight-matrix orthogonality to stabilize learning (Chung et al., 2024), and discouraging excessive feature shifts during training (Moalla et al., 2024). Some approaches balance adaptation and retention by maintaining separate fast and slow learners, periodically resetting the fast learner to the slow one (Lee et al., 2024). Other methods target specific causes, such as activation sparsity reducing gradients (Abbas et al., 2023), while others show that combining mechanisms like layer normalization and weight decay strengthens plasticity preservation (Lyle et al., 2025). Recent studies have also linked data diversity to plasticity preservation: Mayor et al. (2025) improve diversity through multiple parallel environments, and McLean et al. (2025) do so through multi-task training. While these approaches rely on optimizer adaptations, regularization, architectural modifications, or data diversity to preserve plasticity, our method achieves both diverse data collection and plasticity preservation within a single probabilistic framework using evidential value learning. We compare our method with Dohare et al. (2024) and Moalla et al. (2024), which represent state-of-the-art solutions for loss of plasticity in PPO.

**Directed exploration** encourages agents to seek out novel or informative states. Prior work has improved the exploration schemes of PPO in both stationary and non-stationary settings (Burda et al., 2019; Wang et al., 2019; Zhang et al., 2022; Steinparz et al., 2022; Yang et al., 2024b). PPO is also widely applied in continuous control, particularly in robotics (Schwarke et al., 2025; Mittal et al., 2024), where non-stationarity often arises due to dynamic environments and changing task requirements. To address this, Schwarke et al. (2023) extended random network distillation (Burda et al., 2019) to non-stationary robotic tasks, while Yang et al. (2024b) further advanced it with a distributional formulation. We compare against Yang et al. (2024b), which represents the state-of-the-art in directed exploration for PPO, with its non-distributional variant demonstrating strong performance in non-stationary robotics task (Schwarke et al., 2023).

**Relation of non-stationary reinforcement learning to curriculum, meta-, and continual reinforcement learning.** Continual learning aims to learn a single policy that maximizes the expected return in a stream of tasks (Khetarpal et al., 2022). This is a useful learning setup when the tasks have common properties. The challenge is to improve performance in each new task by exploiting these commonalities without causing a performance drop in the previously seen tasks, the phenomenon called *catastrophic forgetting* (Rusu et al., 2016; Kirkpatrick et al., 2017; Traoré et al., 2019; Kaplanis et al., 2019). Non-stationary reinforcement learning addresses setups where some factors of the environment change slower than the step-level transition dynamics. In such setups, the agent is tasked to adapt to the recent change as quickly as possible and does not have to remember all the history of the changes. Instead, it uses all the model capacity to capture the

*trend* of the change and rapidly adapt to it. Our setup can thus be viewed as a special case of a continual reinforcement learning problem with weaker assumptions than standard benchmarks (Wołczyk et al., 2021; Tomilin et al., 2023), such as the agent having no access to task identifiers, task-specific replay buffers, or network heads. These relaxed assumptions make our non-stationary setting an even more challenging instance of continual reinforcement learning. In curriculum learning, task order is typically chosen, either explicitly or implicitly, to promote transfer or accelerate mastery (Bengio et al., 2009). In our case, tasks change at short, fixed intervals, independent of success, difficulty, or skill acquisition. The sequences are structured but not pedagogically curated; they reflect arbitrary shifts in environment dynamics rather than a progression intended to guide the learner. Meta-learning trains agents across task distributions for fast generalization at test time (Al-Shedivat et al., 2018; Berseth et al., 2021; Bing et al., 2023). Methods in these areas often stress retention by constraining updates, which can reduce plasticity. In non-stationary settings, however, plasticity is crucial for effective long-term adaptation with minimal regret. Despite its importance, non-stationary reinforcement learning has been less studied than meta- or continual learning (Khetarpal et al., 2022). Compared to continual learning formulations such as Dohare et al. (2024), our setting differs in two ways: (i) changes occur at shorter intervals, requiring faster adaptation, and (ii) changes are structured rather than random, enabling realistic strategies. In domains like robotics or control under changing conditions, retaining performance across all past dynamics is often infeasible; the main challenge is rapid adaptation to current conditions, highlighting non-stationary reinforcement learning as a distinct and practical setting.

## 2.2 Evidential deep learning

Bayesian inference (Bishop, 2006; Gelman et al., 2013) infers a posterior distribution over model parameters from a given likelihood function evaluated on data and a prior distribution chosen without access to data. Evidential deep learning (Sensoy et al., 2018) applies the classical Bayesian framework in a particular way where posteriors are fit to data-specific random variables from data-specific prior distributions, the parameters of which are amortized by input observations. The amortized prior and the likelihood are chosen from conjugate families to ensure analytically tractable computation of the posterior and the marginal likelihood, the latter of which is used as a training objective.[1] See Ulmer et al. (2023) for a recent survey paper introducing the concept in greater detail.

We build our solution on Amini et al. (2020)'s adaptation of the evidential framework to regression problems, as a typical continuous control task has real-valued reward functions. Amini et al. (2020)'s approach assumes that the output label $y$ corresponding to an input observation $\boldsymbol{x}$ follows a normally distributed likelihood with mean $\mu$ and variance $\sigma^2$. This distribution is assigned a Normal Inverse-Gamma ($\mathcal{NIG}$) distributed evidential prior:

$$(\mu, \sigma^2)|\boldsymbol{m}(\boldsymbol{x}) \sim \mathcal{NIG}\left(\mu, \sigma^2|\omega(\boldsymbol{x}), \nu(\boldsymbol{x}), \alpha(\boldsymbol{x}), \beta(\boldsymbol{x})\right)$$
$$= \mathcal{N}\left(\mu|\omega(\boldsymbol{x}), \sigma^2\nu(\boldsymbol{x})^{-1}\right)\mathcal{I}nv\mathcal{G}am\left(\sigma^2|\alpha(\boldsymbol{x}), \beta(\boldsymbol{x})\right),$$

where the hyperparameters $\omega, \nu, \alpha, \beta$ are modeled as input-dependent functions, specifically neural networks with weights $\phi$. Throughout the paper, we suppress the dependency of the variables on $\phi$ and $\boldsymbol{x}$ for notational clarity, e.g., $\omega = \omega_\phi(\boldsymbol{x})$, and refer to them jointly as the evidential parameters $\boldsymbol{m} \triangleq \boldsymbol{m}_\phi = (\omega, \nu, \alpha, \beta)$. Due to its conjugacy with the normal likelihood $p(y|\mu, \sigma^2) = \mathcal{N}(y|\mu, \sigma^2)$, the posterior $p(\mu, \sigma^2|y, \boldsymbol{m})$ and the marginal likelihood $p(y|\boldsymbol{m})$ are analytically tractable. This marginal is the well-known Student-t distribution:

$$y|\boldsymbol{m} \sim \text{St}\left(y\Big|\omega, \frac{\beta(1+\nu)}{\nu\alpha}, 2\alpha\right).$$

The parameters of this distribution can be fit by minimizing the negative logarithm of the marginal likelihood

$$\mathcal{L}_{\text{NLL}}(\boldsymbol{m}) = \frac{1}{2}\log\left(\frac{\pi}{\nu}\right) - \alpha\log(\Omega) + \left(\alpha + \frac{1}{2}\right)\log\left((y-\omega)^2\nu + \Omega\right) + \log\left(\frac{\Gamma(\alpha)}{\Gamma\left(\alpha + \frac{1}{2}\right)}\right), \qquad (2)$$

where $\Omega = 2\beta(1+\nu)$ and $\Gamma(\cdot)$ is the Gamma function. See Section A for the derivation of the posterior distribution.

---

[1]Marginal likelihood optimization is also known as *Type II Maximum Likelihood* or *Empirical Bayes* (Efron, 2012).

**Evidential deep learning in deep reinforcement learning.** Evidential deep learning has been extensively used in numerous machine learning frameworks and practical tasks (Gao et al., 2024). It has also been integrated into deep reinforcement learning for recommendation systems to provide uncertainty-aware recommendations (Wang et al., 2024), modeling policy network uncertainty to guide evidence-based exploration in behavioral analysis (Wang et al., 2023), incorporating uncertainty measures as rewards for decision-making in opinion inference tasks (Zhao et al., 2019), and calibrating prediction risk in safety-critical vision tasks through fine-grained reward optimization (Yang et al., 2024a). However, we instead use it to model uncertainty in value function estimates, which enables confidence-based exploration and helps preserve the plasticity of the neural network.

## 3 Method

We present a method that adapts the evidential approach to learn a distribution over the value function $V(s_t)$. The inferred distribution induces a corresponding distribution over the GAE, which enables the model to detect distributional shifts resulting from the non-stationarity of the dynamics and to guide directed exploration, thereby promoting rapid adaptation.

### 3.1 Evidential value learning

We assume our value function estimates $V(s_t)$ to be normally distributed with unknown mean $\mu$ and variance $\sigma^2$, which are jointly $\mathcal{NIG}$-distributed. We shorten the notation to $V_t = V(s_t)$ when the relation is clear from context. Although evidential deep learning has demonstrated promising results in epistemic uncertainty estimation, including for unseen out-of-distribution data, naïvely following Amini et al. (2020)'s method often results in training instabilities even on standard supervised regression tasks (Oh & Shin, 2022; Meinert et al., 2023). Prior work relied on non-Bayesian heuristics to solve these instabilities. We are the first to introduce a fully Bayesian hierarchical design to solve this which we summarize in the plate diagram in Figure 2. Introducing hyperpriors on each of the four evidential parameters, our model is:

$$\omega(s) \sim \mathcal{N}\left(\omega(s)|\mu_\omega^0, (\sigma_\omega^0)^2\right),$$
$$\nu(s) \sim \mathcal{G}am\left(\nu(s)|\alpha_\nu^0, \beta_\nu^0\right),$$
$$\alpha(s) \sim \mathcal{G}am\left(\alpha(s)|\alpha_\alpha^0, \beta_\alpha^0\right),$$
$$\beta(s) \sim \mathcal{G}am\left(\beta(s)|\alpha_\beta^0, \beta_\beta^0\right),$$
$$\sigma^2 \sim \mathcal{I}nv\mathcal{G}am\left(\sigma^2|\alpha(s), \beta(s)\right),$$
$$\mu|\sigma^2 \sim \mathcal{N}\left(\mu|\omega(s), \sigma^2\nu(s)^{-1}\right),$$
$$V|\mu, \sigma^2 \sim \mathcal{N}(V|\mu, \sigma^2),$$

Figure 2: Plate diagram of our evidential value learning model.

where $\mathcal{G}am(\cdot)$ is the Gamma distribution, and $\mu_\omega^0, \ldots, \beta_\beta^0$ are fixed hyperparameters.[2] We adopt a fixed set of hyperpriors to provide relatively flat and uninformative priors for all experiments. See Table 12 in the Appendix for further details. Following our notational convention, we suppress the dependency on $s$, e.g., $\omega = \omega(s)$, and combine the evidential parameters into $\boldsymbol{m} = (\omega, \nu, \alpha, \beta)$. Marginalizing over $(\mu, \sigma^2)$ yields

$$p(V, \boldsymbol{m}) = \int p(V|\mu, \sigma^2)p(\mu, \sigma^2|\boldsymbol{m})d(\mu, \sigma^2)\, p(\boldsymbol{m}) = p(V|\boldsymbol{m})p(\boldsymbol{m}),$$

where $p(V|\boldsymbol{m})$ is a Student-t distribution parameterized as in Section 2.2. The hyperprior $p(\boldsymbol{m})$ acts as a regularizer in the log-joint objective. The training objective of evidential value learning is $\mathcal{L}(\boldsymbol{m}) = \mathcal{L}_{\text{NLL}}(\boldsymbol{m}) - \xi \log p(\boldsymbol{m})$, where $\xi \geq 0$ is a regularization coefficient.

---

[2]As $\omega$ is a deterministic transformation of the state $s$, the notation $\omega(s) \sim \mathcal{N}(\cdot)$ implies that its parameters $\phi$ are random variables such that $\omega(s)$ is normally distributed.

The mean and variance of the state-value function output $V$ can be computed analytically as

$$\mathbb{E}_{V|\boldsymbol{m}}[V] = \mathbb{E}_{(\mu,\sigma^2)|\boldsymbol{m}}\left[\mathbb{E}_{V|\mu,\sigma^2}[V]\right] = \mathbb{E}_{(\mu,\sigma^2)|\boldsymbol{m}}[\mu] = \omega,$$

and

$$\begin{aligned}
\mathrm{var}_{V|\boldsymbol{m}}[V] &= \mathbb{E}_{(\mu,\sigma^2)|\boldsymbol{m}}\left[\mathrm{var}_{V|\mu,\sigma^2}[V]\right] + \mathrm{var}_{(\mu,\sigma^2)|\boldsymbol{m}}\left[\mathbb{E}_{V|\mu,\sigma^2}[V]\right] \\
&= \mathbb{E}_{(\mu,\sigma^2)|\boldsymbol{m}}\left[\sigma^2\right] + \mathrm{var}_{(\mu,\sigma^2)|\boldsymbol{m}}[\mu] \\
&= \frac{\beta}{\alpha - 1} + \frac{\beta}{\nu(\alpha - 1)} = \frac{\beta}{\alpha - 1}\left(1 + \frac{1}{\nu}\right),
\end{aligned}$$

where we assume $\alpha > 1$.[3] The first equality follows from the law of total variance, which splits the marginal variance into aleatoric and epistemic uncertainty components. Reliance on $\mathrm{var}_{y|\boldsymbol{m}}[y]$ therefore provides us with a principled way of incorporating irreducible uncertainty inherent in the environmental structure and reducible uncertainty due to improvable approximation errors in EPPO.

**Distributional reinforcement learning.** Evidential value learning belongs to a broader research field that incorporates distributional information into reinforcement learning models, which can be roughly divided into two sub-fields. The first aims to account for aleatoric uncertainty in the MDP caused by the inherent stochasticity of the environment. It focuses on accurately modeling the resulting distribution over the returns $G_t$, e.g., to infer risk-averse policies (Keramati et al., 2020). See Bellemare et al. (2023) for a recent textbook introduction. The second focuses on accounting for second-degree epistemic uncertainty inherent in value function inference, usually relying on methods from Bayesian inference (Ghavamzadeh et al., 2015; Luis et al., 2024), e.g., to use it as a guide for exploration (e.g., Deisenroth & Rasmussen, 2011; Osband et al., 2019). Evidential value learning differs from standard distributional reinforcement learning approaches in several key ways. While methods in the first category focus solely on modeling aleatoric uncertainty for risk-sensitive, usually risk-averse, control, evidential value learning is related to the second area of research. It uses an evidential model over the value function to induce a distribution over an advantage function that simultaneously incorporates both aleatoric and epistemic uncertainty. To make this feasible requires the assumption of a parameterized density model instead of the model-free particle-based distribution approximation used in distributional reinforcement learning. This dual uncertainty quantification enables both regularization benefits and optimistic exploration strategies, distinguishing it from approaches that target only adaptive or risk-sensitive settings through aleatoric uncertainty modeling alone.

### 3.2 Directed exploration via probabilistic advantages

Evidential value learning provides an uncertainty quantifier for the value function that detects shifts in the data distribution caused by non-stationary state transition dynamics. By parameterizing the value function as a distribution $p(V|\boldsymbol{m})$ rather than as a point estimate, our model naturally increases uncertainty in regions of distributional shift, thereby maintaining gradient flows and preserving plasticity. This uncertainty propagates through the advantage calculation, turning the generalized advantage estimator $\hat{A}_t^{\mathrm{GAE}}$ into a random variable. An Upper Confidence Bound (UCB) exploration strategy emerges naturally from this probabilistic treatment, drawing theoretical justification from the exploration-exploitation trade-off in multi-armed bandit theory (Auer et al., 2002). The UCB estimator

$$\hat{A}_t^{\mathrm{UCB}} = \mathbb{E}\left[\hat{A}_t^{\mathrm{GAE}}\right] + \kappa\sqrt{\mathrm{var}\left[\hat{A}_t^{\mathrm{GAE}}\right]}, \tag{3}$$

provides an optimistic estimator that balances expected advantage with uncertainty, where $\kappa > 0$ controls the confidence radius. This ensures that the policy is directed towards state-action regions where the value function exhibits high uncertainty, enabling curiosity-driven exploration with firm theoretical grounding.

The mean estimate for GAE is

$$\mathbb{E}\left[\hat{A}_t^{\mathrm{GAE}}\right] = \sum_{l=0}^{\infty}(\gamma\lambda)^l\mathbb{E}\left[\delta_{t+l}\right],$$

---

[3]We enforce this condition by adding one to the neural network's output.

and remains tractable due to the linearity of expectations and because the mean of the temporal difference $\mathbb{E}[\delta_t] = r_t + \gamma\mathbb{E}[V_{t+1}] - \mathbb{E}[V_t]$ is tractable. We propose two variants of EPPO that differ in how the variance term $\text{var}[\hat{A}_t^{\text{GAE}}]$ in Equation (3) is computed.

**(EPPO$_{\text{cor}}$) Exploration via correlated uncertainties.** We derive the variance of $\hat{A}_t$ by focusing on its definition as the exponentially weighted average of the $k$-step estimators $\hat{A}_t^{(k)} = -V_t + \gamma^k V_{t+k} + \sum_{l=0}^{k-1}\gamma^l r_{t+l}$. Because the rewards are deterministic in our setup and therefore have zero variance, we combine them into a generic constant term and obtain

$$\hat{A}_t^{\text{GAE}} \triangleq (1-\lambda)\sum_{l=1}^{\infty}\lambda^{l-1}\hat{A}_t^{(l)} = (1-\lambda)\left(-V_t\sum_{l=0}^{\infty}\lambda^l + \sum_{l=1}^{\infty}\gamma^l\lambda^{l-1}V_{t+l}\right) + \text{const}$$

$$= -V_t + \frac{1-\lambda}{\lambda}\sum_{l=1}^{\infty}(\gamma\lambda)^l V_{t+l} + \text{const}.$$

Given the assumed conditional independence of the states, the resulting variance is

$$\text{var}\left[\hat{A}_t^{\text{GAE}}\right] = \text{var}[V_t] + \left(\frac{1-\lambda}{\lambda}\right)^2\sum_{l=1}^{\infty}(\gamma\lambda)^{2l}\text{var}[V_{t+l}]. \tag{4}$$

We use this variance to construct the UCB in Equation (3) and refer to it as EPPO$_{\text{cor}}$ in the experiments.

**(EPPO$_{\text{ind}}$) Exploration via uncorrelated uncertainties.** We also consider the case where the $k$-step estimators $\hat{A}_t^{(k)}$ are assumed to be independent of each other. We then construct the overall variance as the exponentially weighted sum of the individual $k$-step estimators. It can be shown easily (see Section A.2) that the resulting variance approximation is

$$\text{var}\left[\hat{A}_t^{\text{GAE}}\right] \approx \frac{1-\lambda}{1+\lambda}\text{var}[V_t] + \left(\frac{1-\lambda}{\lambda}\right)^2\sum_{l=1}^{\infty}(\gamma\lambda)^{2l}\text{var}[V_{t+l}], \tag{5}$$

i.e., the influence of the current value variance is down-scaled by a factor $(1-\lambda)/(1+\lambda) < 1$ relative to the future time steps in EPPO$_{\text{ind}}$ compared with EPPO$_{\text{cor}}$. This adjustment makes EPPO$_{\text{ind}}$ more far-sighted for the same $\kappa$. We use the variance estimate in Equation (5) to construct the UCB in Equation (3).

**Practical implementation of EPPO.** We provide pseudocode in Algorithm 1 illustrating how to implement EPPO variants by overlaying color-coded modifications on top of a standard PPO implementation, where each color corresponds to a specific EPPO variant. As shown, EPPO variants require only minimal changes to PPO with a clipped objective and GAE-based advantage calculation. The key additions include an evidential value estimator, its update rule, and a probabilistic advantage computation with UCB. All other components remain identical to those in PPO.

## 4 Experiments

We design experiments to benchmark EPPO variants against state-of-the-art on-policy deep actor-critic algorithms in non-stationary continuous control environments. To amplify the effect of non-stationarity on model performance, we define tasks over short time intervals and introduce changes in the environment dynamics. In each interval, agents are required to detect the change, explore effectively, and adapt rapidly to maximize the overall return during learning. We focus on non-stationarities that affect environment dynamics in a structured manner, based on identifiable patterns of change, and exclude scenarios where changes occur randomly. This design ensures that observed performance improvements stem from enhanced learning capabilities rather than increased robustness to noise. We run our simulations on the `Ant` and `HalfCheetah` environments using the 'v5' versions of MuJoCo environments (Todorov et al., 2012). For further details on the experimental pipeline and hyperparameters, see Section B. The implementation of the EPPO variants and the full experimental pipeline is available at `https://github.com/adinlab/EPPO`.

Table 1: *Performance evaluation on slippery environments.* Area Under the Learning Curve (AULC) and Final Return (mean±se) scores are averaged over 15 repetitions. The highest averages are highlighted in bold, and results that are not significantly different from them (see Section B.2) are underlined. The average score represents the mean across all environments, while the average ranking is based on the mean scores.

| Metric | Model | decreasing | | increasing | | Average | |
|---|---|---|---|---|---|---|---|
| | | Ant | HalfCheetah | Ant | HalfCheetah | Score | Ranking |
| AULC (↑) | PPO | $2355_{\pm 203}$ | $2495_{\pm 201}$ | $2237_{\pm 254}$ | $2536_{\pm 297}$ | 2406 | 5.3 |
| | PFO | $2522_{\pm 109}$ | $2300_{\pm 189}$ | $2485_{\pm 90}$ | $1809_{\pm 430}$ | 2279 | 5.0 |
| | CB | $2185_{\pm 63}$ | $1519_{\pm 183}$ | $2288_{\pm 118}$ | $1833_{\pm 150}$ | 1956 | 6.5 |
| | PPO$_{DRND}$ | $2475_{\pm 139}$ | $2633_{\pm 180}$ | $2307_{\pm 348}$ | $2021_{\pm 246}$ | 2359 | 4.5 |
| | EPPO$_{mean}$ | $2504_{\pm 127}$ | $2432_{\pm 299}$ | $2875_{\pm 77}$ | $2822_{\pm 219}$ | 2658 | 3.5 |
| | EPPO$_{cor}$ | $2561_{\pm 128}$ | $2699_{\pm 256}$ | $\mathbf{2944_{\pm 80}}$ | $\mathbf{3645_{\pm 240}}$ | **2962** | **1.5** |
| | EPPO$_{ind}$ | $\mathbf{2614_{\pm 138}}$ | $\mathbf{2866_{\pm 218}}$ | $2779_{\pm 86}$ | $3374_{\pm 220}$ | 2908 | 1.8 |
| Final Return (↑) | PPO | $2357_{\pm 230}$ | $2483_{\pm 212}$ | $2341_{\pm 270}$ | $2720_{\pm 310}$ | 2475 | 5.5 |
| | PFO | $2613_{\pm 110}$ | $2346_{\pm 214}$ | $2620_{\pm 99}$ | $1906_{\pm 462}$ | 2371 | 5.3 |
| | CB | $2303_{\pm 72}$ | $1527_{\pm 193}$ | $2398_{\pm 116}$ | $1920_{\pm 172}$ | 2037 | 6.5 |
| | PPO$_{DRND}$ | $2583_{\pm 141}$ | $2672_{\pm 191}$ | $2428_{\pm 368}$ | $2115_{\pm 259}$ | 2449 | 4.5 |
| | EPPO$_{mean}$ | $2660_{\pm 131}$ | $2522_{\pm 331}$ | $3002_{\pm 92}$ | $2978_{\pm 227}$ | 2790 | 3.0 |
| | EPPO$_{cor}$ | $2714_{\pm 128}$ | $2821_{\pm 274}$ | $\mathbf{3071_{\pm 88}}$ | $\mathbf{3872_{\pm 248}}$ | **3120** | **1.5** |
| | EPPO$_{ind}$ | $\mathbf{2741_{\pm 145}}$ | $\mathbf{2970_{\pm 231}}$ | $2941_{\pm 89}$ | $3559_{\pm 227}$ | 3053 | 1.8 |

**Baselines.** To validate our hypothesis that uncertainty-aware value estimation enhances both plasticity preservation and directed exploration, we benchmark against four representative baselines, summarized in Table 2. These baselines are selected to isolate and test the contribution of each hypothesis component. *(i) PPO* (Schulman et al., 2017): A widely used on-policy deep actor-critic reinforcement learning algorithm that serves as the foundation for EPPO. We follow the most recent implementation practices to represent the state of the art. In particular, we use GAE (Schulman et al., 2016) to estimate value function targets. *(ii) PFO* (Moalla et al., 2024): A recent PPO variant that addresses the plasticity problem under non-stationarity by extending the trust region constraint to the feature space. *(iii) CB* (Dohare et al., 2024): A PPO variant for continual learning that mitigates plasticity loss through continual backpropagation, where a fraction of less-used units are randomly reinitialized. *(iv) PPO$_{DRND}$* (Yang et al., 2024b):

Table 2: *Plasticity and exploration.* Comparison of baselines highlighting the presence of plasticity preservation mechanisms and directed exploration.

| Model | Plasticity Mechanism | Directed Exploration |
|---|---|---|
| PPO | ✗ | ✗ |
| PFO | ✓ | ✗ |
| CB | ✓ | ✗ |
| PPO$_{DRND}$ | ✗ | ✓ |
| EPPO$_{mean}$ | ✓ | ✗ |
| EPPO$_{cor}$ | ✓ | ✓ |
| EPPO$_{ind}$ | ✓ | ✓ |

A state-of-the-art PPO variant designed for directed exploration using random network distillation, where the distillation signal acts as a pseudo-count to generate intrinsic rewards that guide exploration. We also evaluate the EPPO variant with $\kappa = 0$, which performs evidential value learning without directed exploration. We refer to this model as *EPPO$_{mean}$*. Its relative performance highlights the contribution of directed exploration. We use the recommended hyperparameters for the baselines from the original papers, except for CB, where we conduct a grid search due to its high sensitivity to two hyperparameters as reported in the original work. We provide the full details in Section B.3.

**Experimental setup.** We propose two experimental setups to assess the ability of the models to adapt to non-stationarity. In both setups, we encourage fast adaptation by limiting task durations to short intervals. We also preserve agent learnability by introducing changes gradually and avoiding abrupt transitions. The setups are as follows:

*(i)* **Slippery environments.** Inspired by Dohare et al. (2021; 2024), we construct non-stationary environments by varying the friction coefficient of the floor in locomotion tasks using the Ant and HalfCheetah

Table 3: *Performance evaluation on paralysis environments.* Area Under the Learning Curve (AULC) and Final Return (mean±se) scores are averaged over 15 repetitions. The highest averages are highlighted in bold, and results that are not significantly different from them (see Section B.2) are underlined. The average score represents the mean across all environments, while the average ranking is based on the mean scores.

| Metric | Environment | Strategy | Model | | | | | | |
|---|---|---|---|---|---|---|---|---|---|
| | | | PPO | PFO | CB | $PPO_{DRND}$ | $EPPO_{mean}$ | $EPPO_{cor}$ | $EPPO_{ind}$ |
| AULC (↑) | Ant | back-one | $2009_{\pm312}$ | $2259_{\pm113}$ | $2298_{\pm121}$ | $2562_{\pm124}$ | $2455_{\pm78}$ | $2608_{\pm129}$ | $\mathbf{2724_{\pm174}}$ |
| | | front-one | $2054_{\pm260}$ | $2098_{\pm87}$ | $2035_{\pm144}$ | $2547_{\pm120}$ | $2407_{\pm92}$ | $\mathbf{2749_{\pm112}}$ | $2743_{\pm121}$ |
| | | back-two | $1928_{\pm174}$ | $2136_{\pm57}$ | $1808_{\pm85}$ | $\mathbf{2226_{\pm92}}$ | $2203_{\pm79}$ | $2099_{\pm80}$ | $2088_{\pm94}$ |
| | | front-two | $1975_{\pm174}$ | $2000_{\pm56}$ | $2023_{\pm78}$ | $2144_{\pm100}$ | $2259_{\pm85}$ | $\mathbf{2294_{\pm93}}$ | $2275_{\pm76}$ |
| | | parallel | $2162_{\pm175}$ | $2298_{\pm86}$ | $1918_{\pm129}$ | $2358_{\pm132}$ | $2350_{\pm103}$ | $2348_{\pm127}$ | $\mathbf{2558_{\pm159}}$ |
| | | cross | $1898_{\pm185}$ | $2161_{\pm71}$ | $1846_{\pm69}$ | $2012_{\pm93}$ | $2167_{\pm91}$ | $2197_{\pm72}$ | $\mathbf{2281_{\pm72}}$ |
| | | Average | 2004 | 2159 | 1988 | 2308 | 2307 | 2383 | **2445** |
| | HalfCheetah | back-one | $2444_{\pm223}$ | $2181_{\pm282}$ | $2258_{\pm142}$ | $2131_{\pm294}$ | $3160_{\pm270}$ | $3502_{\pm173}$ | $\mathbf{3515_{\pm131}}$ |
| | | front-one | $2076_{\pm299}$ | $2485_{\pm271}$ | $2151_{\pm135}$ | $2592_{\pm319}$ | $3384_{\pm227}$ | $3558_{\pm224}$ | $\mathbf{3695_{\pm241}}$ |
| | | cross-v1 | $2311_{\pm235}$ | $2314_{\pm287}$ | $2255_{\pm118}$ | $2487_{\pm245}$ | $3002_{\pm238}$ | $\mathbf{3205_{\pm224}}$ | $3120_{\pm207}$ |
| | | cross-v2 | $2477_{\pm220}$ | $1903_{\pm245}$ | $2318_{\pm95}$ | $2371_{\pm225}$ | $3039_{\pm195}$ | $3250_{\pm195}$ | $\mathbf{3283_{\pm212}}$ |
| | | Average | 2327 | 2221 | 2246 | 2395 | 3146 | 3379 | **3403** |
| | Overall Average | | 2133 | 2184 | 2091 | 2343 | 2643 | 2781 | **2828** |
| | Overall Average Ranking | | 5.9 | 5.2 | 6.2 | 3.8 | 3.1 | 2.1 | **1.7** |
| FINAL RETURN (↑) | Ant | back-one | $2261_{\pm325}$ | $2503_{\pm114}$ | $2478_{\pm126}$ | $2784_{\pm147}$ | $2709_{\pm83}$ | $2891_{\pm129}$ | $\mathbf{2977_{\pm169}}$ |
| | | front-one | $2253_{\pm284}$ | $2337_{\pm87}$ | $2130_{\pm188}$ | $2802_{\pm128}$ | $2649_{\pm96}$ | $\mathbf{3020_{\pm107}}$ | $2956_{\pm135}$ |
| | | back-two | $2188_{\pm205}$ | $2454_{\pm62}$ | $2103_{\pm77}$ | $2512_{\pm91}$ | $\mathbf{2533_{\pm96}}$ | $2400_{\pm86}$ | $2327_{\pm112}$ |
| | | front-two | $2282_{\pm189}$ | $2249_{\pm61}$ | $2310_{\pm92}$ | $2425_{\pm115}$ | $2536_{\pm98}$ | $\mathbf{2633_{\pm97}}$ | $2605_{\pm94}$ |
| | | parallel | $2397_{\pm195}$ | $2601_{\pm95}$ | $2118_{\pm137}$ | $2647_{\pm145}$ | $2649_{\pm114}$ | $2653_{\pm144}$ | $\mathbf{2883_{\pm163}}$ |
| | | cross | $2144_{\pm220}$ | $2467_{\pm79}$ | $2043_{\pm80}$ | $2310_{\pm104}$ | $2495_{\pm103}$ | $2500_{\pm75}$ | $\mathbf{2570_{\pm72}}$ |
| | | Average | 2254 | 2435 | 2197 | 2580 | 2595 | 2683 | **2720** |
| | HalfCheetah | back-one | $2504_{\pm260}$ | $2275_{\pm303}$ | $2344_{\pm152}$ | $2204_{\pm317}$ | $3320_{\pm287}$ | $3696_{\pm178}$ | $\mathbf{3718_{\pm133}}$ |
| | | front-one | $2115_{\pm325}$ | $2577_{\pm295}$ | $2184_{\pm146}$ | $2625_{\pm362}$ | $3540_{\pm235}$ | $3724_{\pm232}$ | $\mathbf{3892_{\pm248}}$ |
| | | cross-v1 | $2405_{\pm271}$ | $2349_{\pm310}$ | $2333_{\pm131}$ | $2648_{\pm246}$ | $3159_{\pm254}$ | $\mathbf{3420_{\pm231}}$ | $3341_{\pm220}$ |
| | | cross-v2 | $2550_{\pm235}$ | $1953_{\pm260}$ | $2434_{\pm106}$ | $2511_{\pm250}$ | $3217_{\pm204}$ | $3450_{\pm208}$ | $\mathbf{3468_{\pm228}}$ |
| | | Average | 2394 | 2288 | 2324 | 2497 | 3309 | 3573 | **3605** |
| | Overall Average | | 2310 | 2376 | 2248 | 2547 | 2881 | 3039 | **3074** |
| | Overall Average Ranking | | 5.8 | 5.3 | 6.4 | 3.9 | 3.9 | 1.9 | **1.7** |

environments. We induce non-stationarity in them by changing friction every 500 000 steps. To create challenging task changes, we implement two strategies: `decreasing`, where friction starts at its maximum value and gradually decreases, and `increasing`, where friction starts at its minimum value and gradually increases. This setup ensures that agents encounter non-stationarity in both increasing and decreasing friction scenarios. The minimum friction is set to 0.5 and the maximum to 4.0, based on the feasibility of solving the tasks—extreme friction values may make movement too difficult due to slipping or an inability to move forward. We define 15 tasks by changing the friction with a positive or negative offset of 0.25.

*(ii)* **Paralysis environments.** We design a new set of non-stationarity experiments by dynamically altering the torque capabilities of the leg joints in the `Ant` and `HalfCheetah` environments, inspired by Al-Shedivat et al. (2018). Each experiment involves paralyzing different joints to diversify the control tasks across experiments. We generate six torque modification schemes for `Ant` and four for `HalfCheetah`. In each scheme, we select specific joints and progressively reduce their torque capability until they become fully paralyzed. Then, we gradually restore their functionality, returning to the fully operational state. This yields a sequence of nine tasks, where each joint either loses or regains 25% of its torque capacity in each step, following the pattern: $[100, 75, 50, 25, 0, 25, 50, 75, 100]$.

**Evaluation metrics.** We assess model performance using two metrics: *(i) Area Under the Learning Curve (AULC)* and *(ii) Final Return. AULC* is the average return collected over the entire training trajectory. It captures not only the agent's final performance but also how quickly and consistently it improves throughout training. In non-stationary environments, where the task dynamics change over time, AULC reflects the agent's ability to continually adapt to new conditions and recover from changes. A higher score indicates

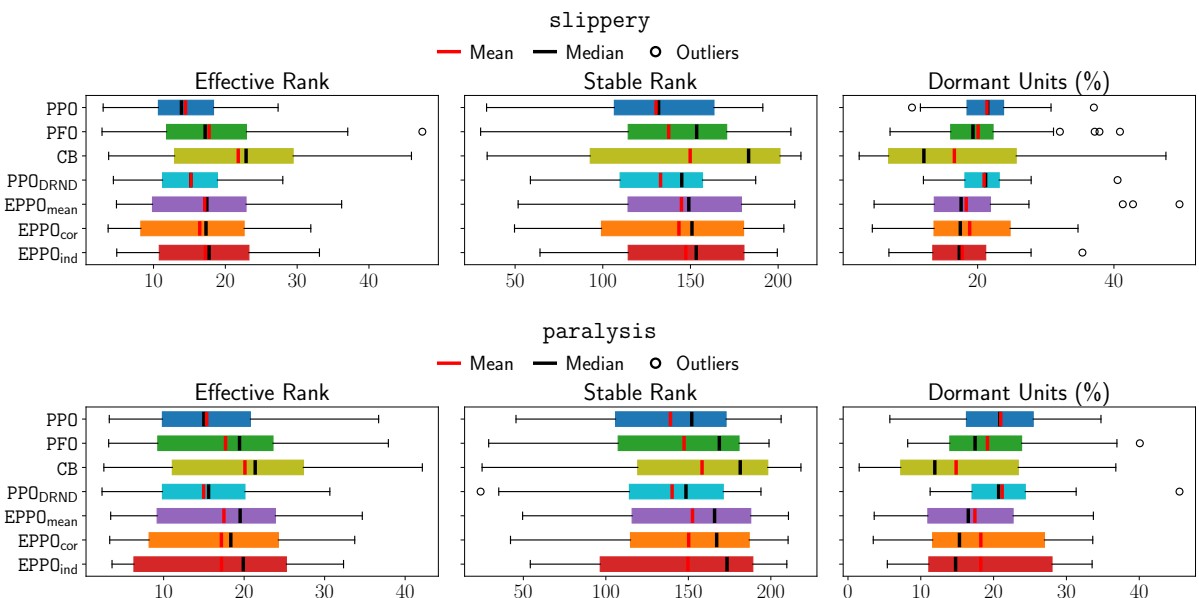

Figure 3: *Plasticity preservation analysis using critic network metrics.* We evaluate three metrics: effective rank, stable rank, and dormant unit percentage, shown from left to right. The top row shows results from the `slippery` environments, and the bottom row shows results from the `paralysis` environments. Each box plot summarizes the distribution of the respective metric across training seeds: the red line indicates the mean, the black line indicates the median, and the individual points represent outliers. These metrics quantify the prediction capacity of the critic networks as learning progresses. EPPO variants consistently preserve plasticity better than PPO variants, as shown by higher ranks and lower dormant unit percentages.

stronger overall adaptation and learning stability across the full training horizon. *Final Return* is calculated at the completion of each individual task by averaging the returns during the last evaluation steps of that task. These evaluations are averaged over all tasks at the end of training. This metric measures how well the agent adapts and performs on individual tasks after it has observed and interacted with them. A higher final return suggests more effective task-specific adaptation. Together, these metrics assess distinct but complementary aspects of adaptation: AULC evaluates an agent's learning efficiency and dynamic adaptability to changing environments over time, while final return assesses the agent's stable performance after task-specific adaptation has occurred. We use statistical tests to identify significant improvements; see Section B.2 for details.

## 4.1 Results and Discussion

We present the detailed results of the experiments in Table 1 and Table 3. We also provide experimental result visualizations in Section B.4, illustrating episode returns throughout the changing tasks and demonstrating both quantitative and qualitative performance differences between EPPO variants and our baselines.

**Plasticity preservation analysis.** We analyze the plasticity of the agents' critic networks using three metrics: *(i) Effective rank* (Roy & Vetterli, 2007) quantifies the number of significant dimensions in the feature matrix. A high value indicates that most matrix dimensions contribute, suggesting that the network generates diverse features and maintains plasticity. *(ii) Stable rank* (Yang et al., 2020) measures the effective dimensionality of the feature matrix. A low rank suggests limited diversity in the learned representations, implying that the network struggles to preserve plasticity. *(iii) Dormant unit percentage* (Sokar et al., 2023) refers to the proportion of inactive neurons. A high percentage suggests impaired gradient flow and reduced learning capacity, indicating a loss of plasticity. We compute these metrics at the end of each task and report the average over the entire training process. Figure 3 summarizes the results across both experimental setups. As shown, EPPO variants consistently achieve higher effective and stable ranks and exhibit fewer dormant

units than plain PPO and its recent variants for handling non-stationarity and directed exploration. Pairwise t-tests on these plasticity metrics confirm that EPPO and its variants significantly outperform plain PPO ($p < 0.05$) , indicating statistical significance; see Section B.2 for details. These findings indicate that the evidential value learning framework helps preserve plasticity and that the addition of directed exploration does not compromise it relative to plain PPO. We provide a full analysis in Section B.1.3.

**Plasticity preservation through evidential value learning.** EPPO preserves plasticity by making the loss (Equation (2)) geometry *(i)* smoother (fewer extrema per unit area) and *(ii)* more convex (less saddle points), which facilitates gradient-based training. Table 4 decomposes the evidential learning objective (see Equation (2)) and reports the sign of the derivative of each loss component with respect to the evidential parameters.

Let $\Delta = |y - \omega|$ (absolute value for simplicity) denote the value function approximation error. The derivatives of the loss term that includes the value function approximation error (see the third row in Table 4) with respect to the evidential parameters have negative signs, decreasing the evidential parameters except $\omega$ which is driven toward the target $y$. Conversely, the derivatives of the other loss terms with respect to the evidential parameters have positive signs, increasing the evidential parameters. As a result, when the value function approximation error drives the evidential parameters downward, the other loss terms penalize this change, balancing the tendencies and constraining the solution in the function space. This regulatory effect smooths the loss geometry.

For a gradient-based method to perform well, the loss geometry should exhibit similar convexity in all dimensions; in other words, the eigenvalues of its Hessian should be close to each other (Belkin, 2021). We examine the gradient of the term that includes the value function approximation error with respect to the neural network parameters, compared with the mean squared error as in plain PPO and other baselines. The gradient term for PPO is $2\Delta\nabla_\theta\omega$, where $\theta$ denotes the network parameters. In contrast, the derivative of Equation (2), considering only the value function approximation error, is $\zeta 2\Delta\nabla_\theta\omega$ where $\zeta = \dfrac{\left(\alpha + \frac{1}{2}\right)\nu}{\nu\Delta^2 + 2\beta(1+\nu)}$. In effect, $\zeta$ adaptively rescales the gradient, reducing the step size when the value function approximation error $\Delta$ is large. This dynamic scaling functions as an effective adaptive learning rate and limits excessively large updates that could move the weights into regions with inactive gradients, which may otherwise create dormant units.

Table 4: *Sign of loss derivatives with respect to evidential parameters.* For each loss term listed in the rows (decomposed from Equation (2)), the table reports the sign of the derivative of the loss with respect to each evidential parameter. '+' and '−' denote positive and negative signs of derivatives, respectively, while '0' indicates no dependency. All evidential-related terms have positive derivatives except in the third row, which includes the value function approximation error term $((y - \omega)^2)$, highlighting their role as regularizers.

| | $|y - \omega|$ | $\nu$ | $\alpha$ | $\beta$ |
|---|---|---|---|---|
| $\frac{1}{2}\log\left(\frac{\pi}{\nu}\right)$ | 0 | + | 0 | 0 |
| $-\alpha\log\left(2\beta\left(1+\nu\right)\right)$ | 0 | + | + | + |
| $\left(\alpha+\frac{1}{2}\right)\log\left(\left(y-\omega\right)^2\nu + 2\beta\left(1+\nu\right)\right)$ | − | − | − | − |
| $\log\left(\frac{\Gamma(\alpha)}{\Gamma\left(\alpha+\frac{1}{2}\right)}\right)$ | 0 | 0 | + | 0 |

As shown in Figures 8 to 9, critics trained with evidential value learning maintain lower gradient norms, higher effective and stable ranks, and fewer dormant units compared to critics trained with mean squared error (plain PPO and PPO$_{\mathrm{DRND}}$). In these baselines, increasing gradient norms causes plasticity loss, leading to lower ranks and more dormant units. Since all models use layer normalization and gradient clipping, as is common in the plasticity-preserving literature (Lyle et al., 2023; 2025), these factors are controlled. Therefore, EPPO's improvements in plasticity are due solely to its evidential value learning loss.

**Exploration analysis.** We evaluate the exploration ability of the agents using the percentage *coverage* metric (Mayor et al., 2025), which quantifies the spatial dispersion of state representations. Specifically, we project the representations into a 2D space and compute the proportion of occupied cells in a uniform grid over this projection. A higher coverage indicates better exploration, meaning that the agent visits a broader portion of the state space. We evaluate this metric for plain PPO, PPO$_{\mathrm{DRND}}$, which serves as the

Table 5: *Exploration analysis.* Percentage coverage (mean$_{\pm\text{se}}$) scores are averaged over 15 repetitions. The highest mean values are highlighted in bold and underlined if they fall within one standard error of the best score.

| | PPO | PPO$_{\text{DRND}}$ | EPPO$_{\text{mean}}$ | EPPO$_{\text{cor}}$ | EPPO$_{\text{ind}}$ |
|---|---|---|---|---|---|
| paralyzed-ant-parallel | $7.51 \pm 0.70$ | $8.12 \pm 0.62$ | $\underline{8.90 \pm 0.74}$ | $\underline{9.27 \pm 0.52}$ | $\mathbf{9.34 \pm 0.68}$ |
| paralyzed-cheetah-back-one | $3.18 \pm 1.21$ | $5.26 \pm 1.45$ | $\underline{12.46 \pm 0.34}$ | $\underline{12.61 \pm 0.30}$ | $\mathbf{12.72 \pm 0.29}$ |

strongest baseline for directed exploration, and the EPPO variants on the paralysis ant environment with the parallel strategy and the paralysis cheetah environment with the back-one strategy. Table 5 reports the percentage coverage results. The EPPO variants consistently achieve higher coverage than both plain PPO and PPO$_{\text{DRND}}$, demonstrating their enhanced ability to explore the state space. In non-stationary environments, agents must gather data from diverse and high-reward regions of the state distribution and continually adapt to the newly visited states through plasticity preservation. Notably, PPO$_{\text{DRND}}$ increases the coverage of plain PPO but does not improve plasticity preservation (see Figure 3), and its adaptation performance in terms of AULC and Final Return remains inferior to that of EPPO variants. The strength of evidential value learning lies in its ability to jointly achieve plasticity preservation and directed exploration within a single probabilistic learning framework. We provide further details in Section B.1.4.

**Discussion.** Our experimental findings are as follows:

*(i) Evidential value learning helps preserve plasticity.* EPPO variants yield higher effective and stable ranks while creating fewer dormant units compared to plain PPO, thereby better preserving the plasticity of the critic networks. This capacity to retain plasticity enables EPPO variants to continue adapting to changing environment dynamics.

*(ii) Directed exploration boosts performance.* EPPO variants with directed exploration (EPPO$_{\text{cor}}$ and EPPO$_{\text{ind}}$) outperform the baselines across all metrics. They also surpass EPPO$_{\text{mean}}$, which uses the mean value function for policy improvement. These results highlight the unique contribution of directed exploration to performance. Notably, the addition of directed exploration also improves the performance of plain PPO, as demonstrated by PPO$_{\text{DRND}}$.

*(iii) Plasticity alone or directed exploration alone is insufficient to overcome non-stationarity.* Baselines focusing solely on preserving plasticity (PFO, CB) or solely on directed exploration (PPO$_{\text{DRND}}$) yield partial improvements but fail to adapt contiunially. This indicates that both properties are necessary for sustained performance.

*(iv) Evidential value learning with directed exploration accelerates convergence and improves training stability while preserving plasticity.* EPPO variants achieve superior task adaptation compared to the baselines, as demonstrated by the final return scores and learning curves. They also converge more rapidly and improve training stability, as supported by higher AULC scores. By modeling value uncertainty, evidential value learning maintains plasticity throughout training.

*(v) Equipping an agent with a mechanism to quantify the uncertainty of the value function enables it to preserve plasticity and explore effectively in the face of non-stationary dynamics.* Our best-performing algorithms, which incorporate uncertainty quantification into the value function, allow agents to maintain plasticity and conduct directed exploration. This facilitates rapid and continual adaptation to non-stationary environments, supporting our key hypothesis.

**Compute time.** We perform our experiments using two computers equipped with GeForce RTX 4090 GPUs, an Intel(R) Core(TM) i7-14700K CPU running at 5.6 GHz, and 96 GB of memory. Our experiments are conducted on these two machines with four parallel seeds. We measure approximately the total wall-clock time for the computation of 15 seeds across all environments at 74.8 hours for PPO, 75 hours for PFO, 78.8 hours for CB, 78.1 hours for PPO$_{\text{DRND}}$, 75.3 hours for EPPO$_{\text{mean}}$, 75.4 hours for EPPO$_{\text{cor}}$, and 75.6 hours

for EPPO$_{\text{ind}}$. The total execution time for all experiments reported in this work is approximately 533 hours, equivalent to 22.2 days on two GPU-supported workstations.

## 5 Limitations and broader impact

We observe EPPO to be sensitive to the choice of some hyperparameters, such as the regularization coefficient $\xi$ and the confidence radius $\kappa$. While this is a common weakness of most deep reinforcement learning algorithms, the effect of the resulting brittleness may be larger in non-stationary environments. We expect that choosing the confidence radius based on a generalization bound, as practiced commonly in bandit research (Li et al., 2010; Srinivas et al., 2010; Kaufmann et al., 2012; Lattimore & Szepesvári, 2020) and increasing the Bayesian modeling hierarchy will make EPPO more robust to hyperparameters. As an on-policy policy-gradient algorithm, EPPO shares similar theoretical properties with other PPO variants. The effect of the evidential learning extension on non-asymptotic convergence is a challenging problem; hence, it requires special investigation. Although our study demonstrates that evidential value learning improves the control of non-stationary systems, we did not investigate whether the quantified uncertainties are calibrated and how strong the correlation is between their calibration and performance. We leave this interesting problem to a separate study. Our results are limited to rigid-body locomotors of a single physics engine, despite covering comprehensive variations of challenging scenarios at non-stationarity levels exceeding those of prior studies. We do not expect extending our results to more tasks to bring any additional insights. We view testing our approach on *physical* robotic systems as the natural next step.

Continuous control of a non-stationary environment is the core problem of building an agentic system on a physical platform. Non-stationarity is the essential element of developing co-adaptive environments where robots and humans learn via bilateral feedback. Such co-adaptation is crucial to ensure human-centric growth of the capabilities of agentic systems of the future. Our work contributes to the responsible AI initiative by facilitating the application of the powerful PPO algorithm to co-adaptive system development.

### Acknowledgments

AA, GB, and MH thank the Carlsberg Foundation for supporting their research under grant number CF21-0250. We also thank all reviewers and the area chair for improving the quality of our work through their thorough reviews and active discussion during the rebuttal phase.

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

# APPENDIX

## A  Derivations

### A.1  Derivations for evidential deep learning

We follow the derivations from Amini et al. (2020), adapting them to our notation whenever necessary.

**Normal inverse-gamma ($\mathcal{NIG}$) distribution**  We use the notation

$$
\begin{aligned}
(\mu, \sigma^2)|\boldsymbol{m} &\sim \mathcal{NIG}\left(\mu, \sigma^2|\omega, \nu, \alpha, \beta\right) \\
&= \mathcal{N}(\mu|\omega, \sigma^2\nu^{-1})\mathcal{I}nv\mathcal{G}am(\sigma^2|\alpha, \beta) \\
&= \frac{\beta^\alpha \sqrt{\nu}}{\Gamma(\alpha)\sqrt{2\pi\sigma^2}} \left(\frac{1}{\sigma^2}\right)^{\alpha+1} \exp\left(-\frac{2\beta + \nu(\omega - \mu)^2}{2\sigma^2}\right),
\end{aligned}
$$

where $\omega \in \mathbb{R}$ and $\nu, \alpha, \beta > 0$. The mean, mode, and variance are given by

$$
\mathbb{E}\left[\mu\right] = \omega, \quad \mathbb{E}\left[\sigma^2\right] = \frac{\beta}{\alpha - 1}, \quad \text{var}\left[\mu\right] = \frac{\beta}{\nu(\alpha - 1)}, \qquad \text{for } \alpha > 1.
$$

The second and third terms correspond to aleatoric and epistemic uncertainty, respectively.

**Model evidence and type II maximum likelihood loss**  We derive the model evidence of an $\mathcal{NIG}$ distribution. We marginalize out $\mu$ and $\sigma$:

$$
\begin{aligned}
p(y|\boldsymbol{m}) &= \int_{(\mu,\sigma^2)} p(y|\mu,\sigma^2)p(\mu,\sigma^2|\boldsymbol{m})d(\mu,\sigma^2) \\
&= \int_{\sigma^2=0}^{\infty} \int_{\mu=-\infty}^{\infty} p\left(y|\mu,\sigma^2\right) p\left(\mu,\sigma^2|\boldsymbol{m}\right) d\mu \; d\sigma^2 \\
&= \int_{\sigma^2=0}^{\infty} \int_{\mu=-\infty}^{\infty} p\left(y|\mu,\sigma^2\right) p\left(\mu,\sigma^2|\omega,\nu,\alpha,\beta\right) d\mu \; d\sigma^2 \\
&= \int_{\sigma^2=0}^{\infty} \int_{\mu=-\infty}^{\infty} \left[\sqrt{\frac{1}{2\pi\sigma^2}} \exp\left(-\frac{(y-\mu)^2}{2\sigma^2}\right)\right] \\
&\qquad \left[\frac{\beta^\alpha \sqrt{\nu}}{\Gamma(\alpha)\sqrt{2\pi\sigma^2}} \left(\frac{1}{\sigma^2}\right)^{\alpha+1} \exp\left(-\frac{2\beta + \nu(\omega - \mu)^2}{2\sigma^2}\right)\right] d\mu \; d\sigma^2 \\
&= \int_{\sigma^2=0}^{\infty} \frac{\beta^\alpha \sigma^{-3-2\alpha}}{\sqrt{2\pi}\sqrt{1+1/\nu}\Gamma(\alpha)} \exp\left(-\frac{2\beta + \frac{\nu(y-\omega)^2}{1+\nu}}{2\sigma^2}\right) d\sigma^2 \\
&= \int_{\sigma=0}^{\infty} \frac{\beta^\alpha \sigma^{-3-2\alpha}}{\sqrt{2\pi}\sqrt{1+1/\nu}\Gamma(\alpha)} \exp\left(-\frac{2\beta + \frac{\nu(y-\omega)^2}{1+\nu}}{2\sigma^2}\right) 2\sigma \; d\sigma \\
&= \frac{\Gamma(1/2 + \alpha)}{\Gamma(\alpha)} \sqrt{\frac{\nu}{\pi}} (2\beta(1+\nu))^\alpha \left(\nu\left(y - \gamma\right)^2 + 2\beta(1+\nu)\right)^{-\left(\frac{1}{2}+\alpha\right)},
\end{aligned}
$$

where $\Gamma(\cdot)$ is the Gamma function. Therefore, the evidence distribution $p(y|\boldsymbol{m})$ is a Student-t distribution, i.e.,

$$
p(y|\boldsymbol{m}) = \text{St}\left(y\Big|\omega, \frac{\beta(1+\nu)}{\nu\alpha}, 2\alpha\right),
$$

which is evaluated at $y$ with location parameter $\omega$, scale parameter $\beta(1 - \nu)/\nu\alpha$, and degrees of freedom $2\alpha$. We can compute the negative log-likelihood (NLL) loss as:

$$
\begin{aligned}
\mathcal{L}_{\text{NLL}}(\boldsymbol{m}) &= -\log p(y|\boldsymbol{m}) \\
&= -\log\left(\text{St}\left(y\middle|\omega, \frac{\beta(1+\nu)}{\nu\alpha}, 2\alpha\right)\right) \\
&= \frac{1}{2}\log\left(\frac{\pi}{\nu}\right) - \alpha\log(\Omega) + \left(\alpha + \frac{1}{2}\right)\log\left((y-\omega)^2\nu + \Omega\right) + \log\left(\frac{\Gamma(\alpha)}{\Gamma\left(\alpha + \frac{1}{2}\right)}\right)
\end{aligned}
$$

where $\Omega = 2\beta(1+\nu)$.

### A.2   Derivations for the generalized advantage estimator

Given the definition of the $k$-step estimator as $\hat{A}_t^{(k)} = -V_t + \gamma^k V_{t+k} + \sum_{l=0}^{k-1} \gamma^l r_{t+l}$, we have that

$$
\text{var}\left[\hat{A}_t^{(k)}\right] = \text{var}\left[V_t\right] + \gamma^{2k}\text{var}\left[V_{t+k}\right].
$$

We adapt our estimator's variance approximation for $\text{EPPO}_{\text{ind}}$ to

$$
\begin{aligned}
\text{var}\left[\hat{A}_t^{\text{GAE}}\right] &\approx (1-\lambda)^2 \sum_{l=1}^{\infty} \lambda^{2(l-1)}\text{var}\left[\hat{A}_t^{(l)}\right] \\
&= (1-\lambda)^2\left(\text{var}\left[V_t\right]\sum_{l=0}^{\infty}\lambda^{2l} + \sum_{l=1}^{\infty}\gamma^{2l}\lambda^{2(l-1)}\text{var}\left[V_{t+l}\right]\right) \\
&= \frac{(1-\lambda)^2}{1-\lambda^2}\text{var}\left[V_t\right] + \left(\frac{1-\lambda}{\lambda}\right)^2\sum_{l=1}^{\infty}(\gamma\lambda)^{2l}\text{var}\left[V_{t+l}\right],
\end{aligned}
$$

i.e., the form we have in (5).

## B   Further details on experiments

### B.1   Experiment Details

In this section, we outline the details and design choices for our experiments and non-stationary environments. We use the `Ant` and `HalfCheetah` environments with the 'v5' versions of MuJoCo (Todorov et al., 2012), as these tasks do not reward the agent for maintaining stability.

#### B.1.1   Slippery environments

Our experimental design is inspired by Dohare et al. (2021; 2024). We construct a non-stationary environment by varying the floor's friction coefficient. Searching for feasible friction values, we set the minimum at 0.5 and the maximum at 4.0. Outside of this range, solving the tasks either becomes infeasible or yields low rewards due to excessive action costs, limited movement, or the agent simply falling.

To introduce variation across tasks while ensuring differences between tasks, we incrementally change the friction by 0.25, resulting in 15 distinct tasks. We implement two strategies for these changes:

- `decreasing`: Friction starts at its maximum value and gradually decreases.

- `increasing`: Friction starts at its minimum value and gradually increases.

These setups ensures that the agents experience non-stationarity in both increasing and decreasing friction scenarios. We implement these changes by modifying the publicly available environment XML files[4][5] to adjust the floor friction coefficients.

### B.1.2   Paralysis environments

We introduce a novel set of non-stationarity experiments by dynamically modifying the torque capabilities of leg joints in the `Ant` and `HalfCheetah` environments, inspired by Al-Shedivat et al. (2018). Specifically, we define six torque modification schemes for `Ant` and four for `HalfCheetah`. Each scheme targets selected joints, progressively reducing their torque capacity until they become completely paralyzed, after which their functionality is gradually restored to the fully operational state. This process results in a sequence of nine tasks, where each joint's torque capacity changes in increments of 25%, following the pattern: $[100, 75, 50, 25, 0, 25, 50, 75, 100]$. Note that while the policy can still output full torques, the applied torque is scaled according to the specified coefficients.

**Paralysis on ant.**   The `Ant` environment consists of four legs and eight joints. We design distinct experiments by paralyzing different joints, ensuring that control tasks remain unique across experiments. For instance, if we paralyze the right back leg, we do not conduct a separate experiment on the left back leg, as the locomotion is symmetric and would result in an equivalent control task. We create the following experiments:

- `back-one`: Paralyzing a single back leg. The affected joints are 6 and 7.

- `front-one`: Paralyzing a single front leg. The affected joints are 2 and 3.

- `back-two`: Paralyzing both back legs. The affected joints are $0, 1, 6$, and 7.

- `front-two`: Paralyzing both front legs. The affected joints are $2, 3, 4$, and 5.

- `cross`: Paralyzing diagonally opposite legs (right back and left front). The affected joints are $0, 1, 2$, and 3.

- `parallel`: Paralyzing the left-side legs (one back and one front). The affected joints are $2, 3, 6$, and 7.

**Paralysis on halfCheetah.**   The `HalfCheetah` environment consists of two legs and four joints. To prevent the agent from resorting to crawling, we modify only one joint per leg. We create the following experiments:

- `back-one`: Paralyzing a single joint in the back leg. The affected joint is 2.

- `front-one`: Paralyzing a single joint in the front leg. The affected joint is 5.

- `cross-v1`: Paralyzing diagonally opposite joints in the back and front legs. The affected joints are 2 and 4.

- `cross-v2`: Paralyzing a different pair of diagonally opposite joints in the back and front legs. The affected joints are 1 and 5.

### B.1.3   Plasticity preservation analysis

We calculate the metrics as follows:

---

[4]https://github.com/Farama-Foundation/Gymnasium/blob/main/gymnasium/envs/mujoco/assets/ant.xml
[5]https://github.com/Farama-Foundation/Gymnasium/blob/main/gymnasium/envs/mujoco/assets/half_cheetah.xml

- *Effective rank* is calculated using the feature matrix $\Phi \in \mathbb{R}^{n \times m}$ from the penultimate layer, with singular values $\sigma_k$ for $k = 1, 2, \ldots, q$, where $q = \max(n, m)$. We define $p_k = \dfrac{\sigma_k}{||\boldsymbol{\sigma}||_1}$, where $||\boldsymbol{\sigma}||_1 = \sum_k |\sigma_k|$. The effective rank of $\Phi$ is given by:

$$\text{effective rank}(\Phi) \triangleq \exp H(p_1, p_2, \ldots, p_q),$$

where the entropy $H(p_1, p_2, \ldots, p_q) = -\sum_k p_k \log p_k$.

- *Stable rank* is also computed using the feature matrix $\Phi$ from the penultimate layer, with:

$$\text{stable rank}(\Phi) \triangleq \min_k \left\{ \frac{\sum_i^k \sigma_i^2}{\sum_j^q \sigma_j^2} > 1 - \delta \right\},$$

where $\delta$ is a threshold. We set $\delta = 0.01$, meaning that the selected rank captures at least 99% of the total variance.

- *Dormant unit percentage* measures the portion of neurons that remain consistently inactive across a batch of inputs. We compute activations immediately after applying the nonlinearity and consider a neuron dormant if its output remains below a small threshold (0.01) for all samples in the batch.

We measure these metrics at the end of each task in the evaluation environment and report their averages over the entire training process. The results for each environment are presented in Figures 5 to 7. We also provide the evolution of these metrics along with the maximum absolute gradient of the critic throughout training in Figures 8 to 9.

**Significance test.** Figure 4 presents p-values for the relative performance of the methods, see Section B.2 for more details. Only the lower triangular part of each matrix is shown, where each entry corresponds to the p-value from comparing the row and column methods. The results show that our EPPO variants preserve plasticity significantly better than plain PPO at a 0.05 significance level. Note that each pairwise test is evaluated and reported independently, i.e., no Bonferroni correction was applied.

### B.1.4 Exploration analysis

Following Mayor et al. (2025), we applied UMAP with default parameters to project the states into a 2D space.[6] This projection is necessary due to the large number and high dimensionality of the states. We then applied min–max normalization with a small epsilon ($1e{-}5$) to scale the projected data to the range $[0, 1)$.

Let the resulting projection be denoted as $\mathcal{D} = \{(x_i, y_i) \mid i = 1, \ldots, N\}$, where $x_i, y_i \in [0, 1)$. We discretize the 2D space into a uniform grid of $G \times G$ cells and assign each point $(x_i, y_i)$ to its corresponding grid cell:

$$\text{idx}_i^x = \min\left(\lfloor x_i \cdot G \rfloor, G - 1\right), \quad \text{idx}_i^y = \min\left(\lfloor y_i \cdot G \rfloor, G - 1\right)$$

The set of occupied cells is defined as

$$\mathcal{O} = \left\{(\text{idx}_i^x, \text{idx}_i^y) \mid i = 1, \ldots, N\right\},$$

and the percentage coverage is computed as

$$\text{Percentage Coverage} = \frac{|\mathcal{O}|}{G^2} \times 100$$

where $|\mathcal{O}|$ is the number of unique occupied cells, and $G^2$ is the total number of grid cells.

We used a grid size of $G = 100$, corresponding to a 0.01 resolution per grid cell. Since Mayor et al. (2025) does not report the specific choices of UMAP parameters, scaling method, or grid size, we selected these values to ensure fairness and reproducibility. The same 15 random seeds are used for this experiment as in the other experiments.

---

[6]We relied on the following implementation: https://docs.rapids.ai/api/cuml/stable/api/#umap.

Table 6: $p$-values for effective rank in `slippery`.

|  | PPO | PFO | CB | PPO$_{\text{DRND}}$ | EPPO$_{\text{mean}}$ | EPPO$_{\text{cor}}$ | EPPO$_{\text{ind}}$ |
|---|---|---|---|---|---|---|---|
| PPO |  |  |  |  |  |  |  |
| PFO | 0.020 |  |  |  |  |  |  |
| CB | 0.000 | 0.000 |  |  |  |  |  |
| PPO$_{\text{DRND}}$ | 0.127 | 0.994 | 1.000 |  |  |  |  |
| EPPO$_{\text{mean}}$ | 0.000 | 0.690 | 1.000 | 0.019 |  |  |  |
| EPPO$_{\text{cor}}$ | 0.010 | 0.877 | 1.000 | 0.070 | 0.825 |  |  |
| EPPO$_{\text{ind}}$ | 0.001 | 0.683 | 1.000 | 0.011 | 0.457 | 0.135 |  |

Table 7: $p$-values for effective rank in `paralysis`.

|  | PPO | PFO | CB | PPO$_{\text{DRND}}$ | EPPO$_{\text{mean}}$ | EPPO$_{\text{cor}}$ | EPPO$_{\text{ind}}$ |
|---|---|---|---|---|---|---|---|
| PPO |  |  |  |  |  |  |  |
| PFO | 0.000 |  |  |  |  |  |  |
| PFO | 0.000 | 0.000 |  |  |  |  |  |
| PPO$_{\text{DRND}}$ | 0.747 | 1.000 | 1.000 |  |  |  |  |
| EPPO$_{\text{mean}}$ | 0.000 | 0.621 | 1.000 | 0.000 |  |  |  |
| EPPO$_{\text{cor}}$ | 0.001 | 0.809 | 1.000 | 0.000 | 0.751 |  |  |
| EPPO$_{\text{ind}}$ | 0.003 | 0.789 | 1.000 | 0.000 | 0.736 | 0.486 |  |

Table 8: $p$-values for stable rank in `slippery`.

|  | PPO | PFO | CB | PPO$_{\text{DRND}}$ | EPPO$_{\text{mean}}$ | EPPO$_{\text{cor}}$ | EPPO$_{\text{ind}}$ |
|---|---|---|---|---|---|---|---|
| PPO |  |  |  |  |  |  |  |
| PFO | 0.041 |  |  |  |  |  |  |
| CB | 0.000 | 0.003 |  |  |  |  |  |
| PPO$_{\text{DRND}}$ | 0.211 | 0.884 | 0.999 |  |  |  |  |
| EPPO$_{\text{mean}}$ | 0.000 | 0.044 | 0.874 | 0.002 |  |  |  |
| EPPO$_{\text{cor}}$ | 0.000 | 0.045 | 0.938 | 0.002 | 0.668 |  |  |
| EPPO$_{\text{ind}}$ | 0.000 | 0.004 | 0.702 | 0.000 | 0.248 | 0.051 |  |

Table 9: $p$-values for stable rank in `paralysis`.

|  | PPO | PFO | CB | PPO$_{\text{DRND}}$ | EPPO$_{\text{mean}}$ | EPPO$_{\text{cor}}$ | EPPO$_{\text{ind}}$ |
|---|---|---|---|---|---|---|---|
| PPO |  |  |  |  |  |  |  |
| PFO | 0.000 |  |  |  |  |  |  |
| CB | 0.000 | 0.000 |  |  |  |  |  |
| PPO$_{\text{DRND}}$ | 0.323 | 0.999 | 1.000 |  |  |  |  |
| EPPO$_{\text{mean}}$ | 0.000 | 0.016 | 0.993 | 0.000 |  |  |  |
| EPPO$_{\text{cor}}$ | 0.000 | 0.081 | 0.999 | 0.000 | 0.909 |  |  |
| EPPO$_{\text{ind}}$ | 0.000 | 0.124 | 1.000 | 0.000 | 0.924 | 0.606 |  |

Table 10: $p$-values for dormant unit percentage in `slippery`.

|  | PPO | PFO | CB | PPO$_{\text{DRND}}$ | EPPO$_{\text{mean}}$ | EPPO$_{\text{cor}}$ | EPPO$_{\text{ind}}$ |
|---|---|---|---|---|---|---|---|
| PPO |  |  |  |  |  |  |  |
| PFO | 0.097 |  |  |  |  |  |  |
| CB | 0.001 | 0.004 |  |  |  |  |  |
| PPO$_{\text{DRND}}$ | 0.278 | 0.847 | 0.998 |  |  |  |  |
| EPPO$_{\text{mean}}$ | 0.003 | 0.071 | 0.921 | 0.010 |  |  |  |
| EPPO$_{\text{cor}}$ | 0.003 | 0.089 | 0.973 | 0.010 | 0.698 |  |  |
| EPPO$_{\text{ind}}$ | 0.000 | 0.006 | 0.826 | 0.000 | 0.286 | 0.044 |  |

Table 11: $p$-values for dormant unit percentage in `paralysis`.

|  | PPO | PFO | CB | PPO$_{\text{DRND}}$ | EPPO$_{\text{mean}}$ | EPPO$_{\text{cor}}$ | EPPO$_{\text{ind}}$ |
|---|---|---|---|---|---|---|---|
| PPO |  |  |  |  |  |  |  |
| PFO | 0.000 |  |  |  |  |  |  |
| CB | 0.000 | 0.000 |  |  |  |  |  |
| PPO$_{\text{DRND}}$ | 0.663 | 1.000 | 1.000 |  |  |  |  |
| EPPO$_{\text{mean}}$ | 0.000 | 0.001 | 1.000 | 0.000 |  |  |  |
| EPPO$_{\text{cor}}$ | 0.000 | 0.043 | 1.000 | 0.000 | 0.962 |  |  |
| EPPO$_{\text{ind}}$ | 0.000 | 0.044 | 1.000 | 0.000 | 0.951 | 0.494 |  |

Figure 4: Each lower-triangular entry shows the $p$-value of a one-sided paired $t$-test testing whether the method in the row significantly outperformed the method in the column (see Section B.2). Smaller $p$-values indicate stronger evidence in favor of the row method's superior performance.

## B.2 Statistical significance

Throughout our experiments we evaluate the significance of performance improvements via statistical tests. Our approach performs a one-sided paired t-test between the model with the highest average performance over all seeds against each of the other models. We compare the null hypothesis of equal means compared to the alternative of better performance by the other model. We consider the null hypothesis to be rejected for p-values smaller than 0.05. Runs are paired over random seeds and we assume approximately normally distributed performance values, fulfilling the t-test assumptions. As this procedure only compares baselines against the best-performing method instead of all pairwise combinations and relies on a conservative one-sided testing approach, we do not employ further multiple testing corrections.

## B.3 Hyperparameters

In this section, we provide all the necessary details to reproduce EPPO. We evaluate EPPO with 15 repetitions using the following seeds: $[1, 2, 3, 4, 5, 6, 7, 8, 9, 10, 11, 12, 13, 14, 15]$. Our implementation will be made public upon acceptance. We list the hyperparameters for the experimental pipeline in Table 12.

### B.3.1 Training

**Architecture and optimization details.** We train EPPO for 500 000 steps per task, performing updates to the policy and critic 10 times every 2048 step with a batch size of 256. The learning rate is set to 0.0003 for both the actor and critic, optimized using Adam (Kingma & Ba, 2015). The actor and critic networks each consist of a 2-layer feedforward neural network with 256 hidden units. Unlike other baselines, our critic network outputs four values instead of one to predict the evidential priors. We apply Layer Normalization (Ba et al., 2016) and ReLU activations (Nair & Hinton, 2010) for both networks. The policy follows a diagonal normal distribution. Following common practice in the literature, we set the discount factor to $\gamma = 0.99$, the GAE parameter to $\lambda = 0.95$, and the clipping rate to $\epsilon = 0.2$. Gradient norms are clipped at 0.5, and GAE advantage estimates are normalized within each batch.

**Evaluation details.** We evaluate the models at the beginning and final steps of each task, as well as every 20 000 steps, using 10 evaluation episodes. The evaluation environment seeds are set to the training seed plus 100. For metric calculation, we use the mean return across the evaluation episodes.

**EPPO details.** We set the regularization coefficient ($\xi$) to 0.01 to scale it down, selecting this value heuristically based on its contribution to the total loss. To prevent overfitting and allow flexibility in learning, we use uninformative, flat priors for the hyperprior distributions. Specifically, we choose a normal distribution $\mathcal{N}(\omega|0, 100^2)$ for $\omega$, though a positively skewed distribution may further improve performance. For $\nu$, $\alpha$, and $\beta$, we use a gamma distribution $\mathcal{G}am(5, 1)$ to ensure positivity. Additionally, we shift the hyperprior distribution of $\alpha$ by $+1$ to ensure a finite mean.

**PFO details.** We followed the original implementation and set the feature regularization coefficient to 10.0. No additional hyperparameters were tuned, and the rest of the implementation was kept identical to our PPO setup.

**CB details.** We set the decay rate to 0.0001 and apply $\ell_2$ regularization with a coefficient of 0.0001, following the original paper. We performed a grid search over maturity thresholds in $[1000, 10\,000]$ and replacement rates in $[0.0001, 0.00001, 0.000001]$ using one representative task from each experimental setting. For slippery environments under the increasing setting, we selected a maturity threshold of 1000 with replacement rates of 0.0001 for Ant and 0.00001 for HalfCheetah. For paralysis environments under the back-one setting, we used a maturity threshold of 1000 with replacement rate 0.0001 for Ant, and a maturity threshold of 10 000 with replacement rate 0.00001 for HalfCheetah. We adopted the generate-and-test process from the official repository[7], while keeping the rest of the implementation identical to our PPO setup. We

---

[7] https://github.com/shibhansh/loss-of-plasticity/tree/63c35f3c758bbb713dd42c72d43dc192fde0d109

Table 12: Hyperparameters used in the experimental pipeline.

| Policy learning | |
|---|---|
| Seeds | $[1, 2, \ldots, 15]$ |
| Number of steps per task | 500 000 |
| Learning rate for actor and critic | 0.0003 |
| Horizon | 2048 |
| Number of epochs | 10 |
| Minibatch size | 256 |
| Clip rate $\epsilon$ | 0.2 |
| GAE parameter $\lambda$ | 0.95 |
| Hidden dimensions of actor and critic | $[256, 256]$ |
| Activation functions of actor and critic | ReLU |
| Normalization layers of actor and critic | Layer Norm |
| Optimizer for actor and critic | Adam |
| Discount factor $\gamma$ | 0.99 |
| Maximum gradient norm | 0.5 |
| **Evaluation-related** | |
| Evaluation frequency (steps) | 20 000 and end of the tasks |
| Evaluation episodes | 10 |
| **EPPO-related** | |
| Regularization coefficient ($\xi$) | 0.01 |
| Hyperprior distribution of $w$ | $\mathcal{N}\left(\omega \mid 0, 100^2\right)$ |
| Hyperprior distribution of $\nu$ | $\mathcal{G}am\left(\nu \mid 5, 1\right)$ |
| Hyperprior distribution of $\alpha$ | $\mathcal{G}am\left(\alpha \mid 5, 1\right) + 1^{\dagger}$ |
| Hyperprior distribution of $\beta$ | $\mathcal{G}am\left(\beta \mid 5, 1\right)$ |
| **Grid Search-related** | |
| Seeds | $[1001, 1002, 1003]$ |
| Radius parameter $\kappa$ for $\mathrm{EPPO_{cor}}$ | $[0.01, 0.1, 0.25]$ |
| Radius parameter $\kappa$ for $\mathrm{EPPO_{ind}}$ | $[0.01, 0.05, 0.1]$ |
| Maturity threshold for CB | $[1000, 10\,000]$ |
| Replacement rate for CB | $[0.0001, 0.00001, 0.000001]$ |
| **PFO-related** | |
| Feature regularization coefficient | 10.0 |
| **CB-related** | |
| Regularization rate ($\ell2$) | 0.0001 |
| Decay rate | 0.0001 |
| **$\mathrm{PPO_{DRND}}$-related** | |
| Hidden dimensions of bonus ensemble | $[256, 256, 256, 32]$ |
| Activation functions of bonus ensemble | ReLU |
| Normalization layers of bonus ensemble | None |
| Learning rate for bonus ensemble | 0.0003 |
| Optimizer for bonus ensemble | Adam |
| Number of ensemble elements | 10 |
| Bonus scaling factor | 0.9 |

$^{\dagger}$The +1 ensures a finite mean for $\alpha$.

Table 13: Radius parameters ($\kappa$) of EPPO.

| Experiment | Environment | Strategy | Confidence radius parameter ($\kappa$) | |
| | | | $\text{EPPO}_{\text{cor}}$ | $\text{EPPO}_{\text{ind}}$ |
|---|---|---|---|---|
| Slippery | Ant | decreasing | 0.05 | 0.1 |
| | | increasing | 0.1 | 0.25 |
| | HalfCheetah | decreasing | 0.05 | 0.1 |
| | | increasing | 0.1 | 0.1 |
| Paralysis | Ant | back-one | 0.05 | 0.01 |
| | | front-one | 0.1 | 0.1 |
| | | back-two | 0.01 | 0.25 |
| | | front-two | 0.1 | 0.01 |
| | | cross | 0.01 | 0.1 |
| | | parallel | 0.05 | 0.01 |
| | HalfCheetah | back-one | 0.05 | 0.01 |
| | | front-one | 0.1 | 0.1 |
| | | cross-v1 | 0.05 | 0.25 |
| | | cross-v2 | 0.05 | 0.1 |

observed that the performance of CB is highly sensitive to the choice of maturity threshold and replacement rate but also to random seeds.

**PPO$_{\text{DRND}}$ details.** We use a bonus ensemble with 10 neural networks, each composed of a 4-layer feedforward architecture with hidden dimensions $[256, 256, 256, 32]$. Each network takes a concatenated state-action pair as input. We apply ReLU activations after each layer and do not include normalization layers. We optimize the ensemble using Adam (Kingma & Ba, 2015) with a learning rate of 0.0003. We scale the output of the ensemble by a bonus factor of 0.9 to guide exploration during training. We adopt the architecture and hyperparameters from the original implementation by Yang et al. (2024b).

**Grid search details for $\kappa$ of EPPO.** We introduce a confidence radius parameter ($\kappa$) that controls the level of optimism incorporated into exploration. To determine an appropriate value, we perform a grid search over $\kappa \in [0.01, 0.05, 0.1]$ for $\text{EPPO}_{\text{ind}}$ and $\kappa \in [0.01, 0.1, 0.25]$ for $\text{EPPO}_{\text{cor}}$, selecting these ranges based on their influence on the advantage estimate. We train models using three seeds $(1001, 1002, 1003)$ and exclude them from the main results. After evaluating the AULC metric, we select the optimal $\kappa$ values and use them for EPPO's final evaluation. Table 13 presents the $\kappa$ values selected for training.

**Practical implementation of EPPO.** We provide pseudocode in Algorithm 1 illustrating how to implement EPPO variants by overlaying color-coded modifications on top of a standard PPO implementation, where each color corresponds to a specific EPPO variant. All lines are shared across EPPO variants and PPO unless otherwise indicated in the comment section.

### B.4 Result visualizations

The learning curves across environment steps are illustrated in Figures 10 to 12. In these figures, the thick curve (dashed, dotted, dash-dotted, or solid) represents the mean returns across ten evaluation episodes and 15 random seeds, with the shaded area indicating one standard error from the mean. The legend provides the mean and standard error for the AULC and final return scores, listed in this order. The vertical black dotted lines mark the task changes.

**Algorithm 1** Evidential Proximal Policy Optimization variants (EPPO$_{\text{mean}}$, EPPO$_{\text{cor}}$, EPPO$_{\text{ind}}$) over PPO

1: **Input:** Initial policy parameters $\theta$, value function parameters $\phi$, clipping threshold $\epsilon$, minibatch size $M$, number of update epochs $K$, trajectory horizon $T$, discount factor $\gamma$, GAE parameter $\lambda$, learning rates $\lambda_\pi, \lambda_V$, radius $\kappa$ for EPPO$_{\text{cor}}$ and EPPO$_{\text{ind}}$, regularization coefficient $\xi$

2: **for** each epoch **do**

3:      Roll out policy in the environment and fill the buffer $D$ with $(s_t, a_t, r_t, s_{t+1}, d_t)$

4:      $V_t \leftarrow V_\phi(s_t)$ and $V_{t+1} \leftarrow V_\phi(s_{t+1})$                 ▷ Value estimates for PPO

5:      $\omega_t, \nu_t, \alpha_t, \beta_t \leftarrow V_\phi(s_t)$ and $\omega_{t+1}, \nu_{t+1}, \alpha_{t+1}, \beta_{t+1} \leftarrow V_\phi(s_{t+1})$      ▷ For EPPOs

6:      $V_t \leftarrow \omega_t, \ \ V_{t+1} \leftarrow \omega_{t+1}$             ▷ Mean value estimates of EPPOs

7:      $\text{var}\left[V_t\right] \leftarrow \dfrac{\beta_t}{\alpha_t - 1}\left(1 + \dfrac{1}{\nu_t}\right), \ \ \text{var}\left[V_{t+1}\right] \leftarrow \dfrac{\beta_{t+1}}{\alpha_{t+1} - 1}\left(1 + \dfrac{1}{\nu_{t+1}}\right)$    ▷ Variance value estimates of EPPOs

8:      Compute deltas: $\delta_t \leftarrow r_t + \gamma V_{t+1}(1 - d_t) - V_t$

9:      Initialize accumulator $A \leftarrow 0$ and $\hat{A}$ as empty list of size $T$ for mean

10:      Initialize accumulator $\text{var}\left[A\right] \leftarrow 0$ and $\text{var}\left[\hat{A}\right]$ as empty list of size $T$ for variance ▷ EPPO$_{\text{cor}}$, EPPO$_{\text{ind}}$

11:      **for** $t = T - 1$ to $0$ **by** $-1$ **do**

12:          $A \leftarrow \delta_t + \gamma\lambda A(1 - d_t)$

13:          $\hat{A}_t \leftarrow A$

14:          $\text{var}\left[A\right] \leftarrow (\gamma\lambda)^2 \left(\text{var}\left[V_{t+1}\right] + (1 - d_t)\text{var}\left[A\right]\right)$           ▷ EPPO$_{\text{cor}}$, EPPO$_{\text{ind}}$

15:          $\text{var}\left[\hat{A}_t\right] \leftarrow \text{var}\left[V_t\right] + \left(\dfrac{1 - \lambda}{\lambda}\right)^2 \text{var}\left[A\right]$          ▷ Equation (4) EPPO$_{\text{cor}}$

16:          $\text{var}\left[\hat{A}_t\right] \leftarrow \dfrac{1 - \lambda}{1 + \lambda}\text{var}\left[V_t\right] + \left(\dfrac{1 - \lambda}{\lambda}\right)^2 \text{var}\left[A\right]$        ▷ Equation (5) EPPO$_{\text{ind}}$

17:      **end for**

18:      $\hat{A} \leftarrow \hat{A} + \kappa\sqrt{\text{var}\left[\hat{A}\right]}$ for all samples             ▷ UCB with Equation (3) EPPO$_{\text{cor}}$, EPPO$_{\text{ind}}$

19:      Compute returns: $\hat{R}_t \leftarrow \hat{A}_t + V_t$ and normalize advantages

20:      **for** $k = 1$ to $K$ **do**

21:          Shuffle $D$ and split into minibatches of size $M$

22:          **for** each minibatch **do**

23:              Compute importance ratio: $r_t(\theta) = \dfrac{\pi_\theta(a_t|s_t)}{\pi_{\theta_{\text{old}}}(a_t|s_t)}$

24:              Compute clipped objective: $\mathcal{L}_{\text{clip}}(\theta) = \frac{1}{M}\sum \min\left(r_t(\theta)\hat{A}_t, \text{clip}(r_t(\theta), 1 - \epsilon, 1 + \epsilon)\hat{A}_t\right)$

25:              Update policy with gradient clipping: $\theta \leftarrow \theta + \lambda_\pi\nabla_\theta\mathcal{L}_{\text{clip}}(\theta)$

26:              Compute value loss: $\mathcal{L}_{VF}(\phi) = \frac{1}{T}\sum\left(V_\phi(s_t) - \hat{R}_t\right)^2$          ▷ PPO

27:              Estimate $\omega_t, \nu_t, \alpha_t, \beta_t \leftarrow V_\phi(s_t)$            ▷ EPPOs

28:              Compute model-fit loss: where $\Omega_t = 2\beta_t(1 + \nu_t)$          ▷ Equation (2) EPPOs

$$\mathcal{L}_{\text{NLL}}(\phi) = \frac{1}{T}\sum \frac{1}{2}\log\left(\frac{\pi}{\nu_t}\right) - \alpha_t\log\left(\Omega_t\right) + \left(\alpha_t + \frac{1}{2}\right)\log\left(\left(\hat{R}_t - \omega_t\right)^2\nu_t + \Omega_t\right) + \log\left(\frac{\Gamma\left(\alpha_t\right)}{\Gamma\left(\alpha_t + \frac{1}{2}\right)}\right)$$

29:              Compute regularization: $\mathcal{L}_{reg}(\phi) = \frac{1}{T}\sum \log p(\omega_t) + \log p(\nu_t) + \log p(\alpha_t) + \log p(\nu_t)$    ▷ EPPOs

30:              Compute value loss: $\mathcal{L}_{VF}(\phi) = \mathcal{L}_{\text{NLL}}(\phi) - \xi\mathcal{L}_{reg}(\phi)$          ▷ EPPOs

31:              Update value function with gradient clipping: $\phi \leftarrow \phi - \lambda_V\nabla_\phi\mathcal{L}_{VF}(\phi)$

32:          **end for**

33:      **end for**

34: **end for**

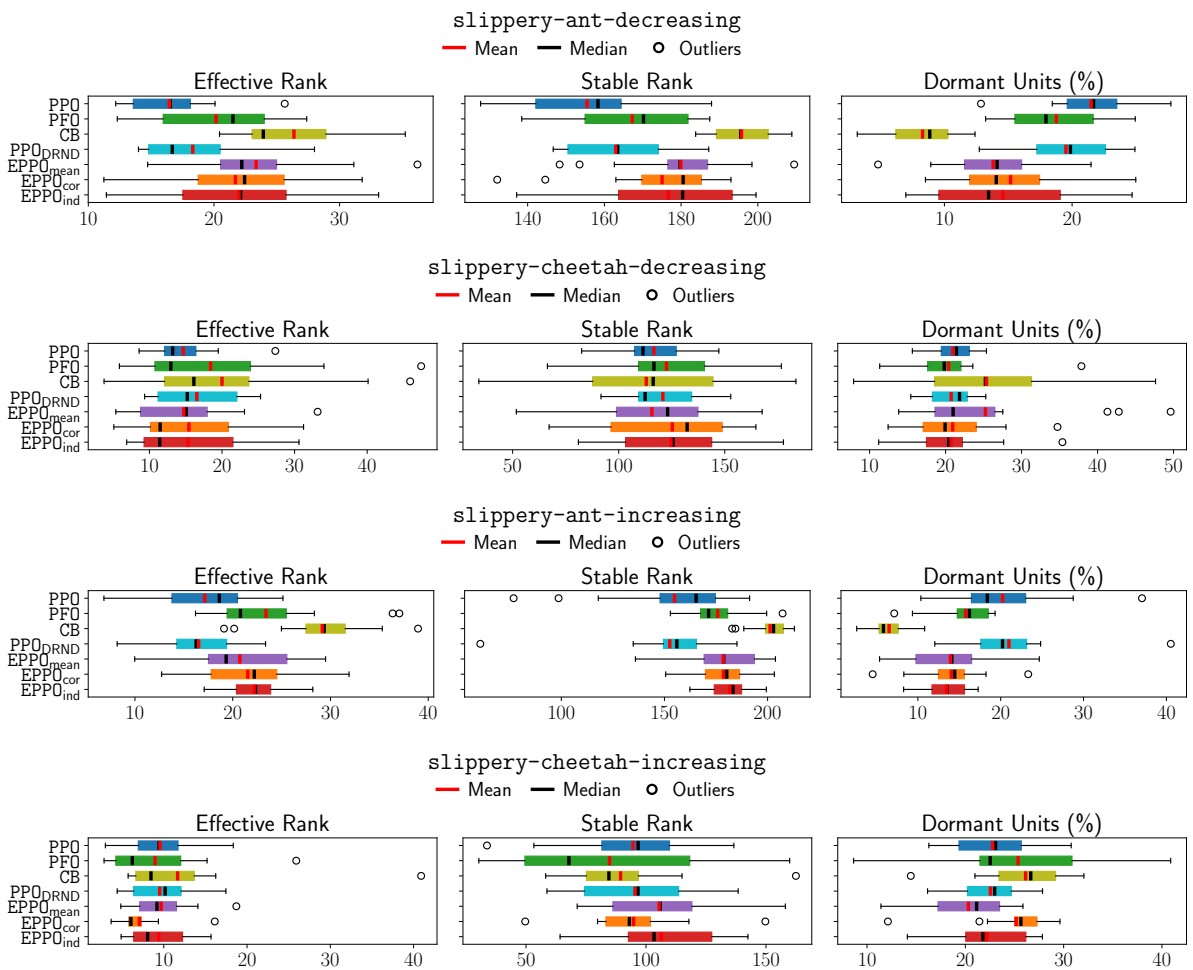

Figure 5: Plasticity preservation analysis for the slippery experiment.

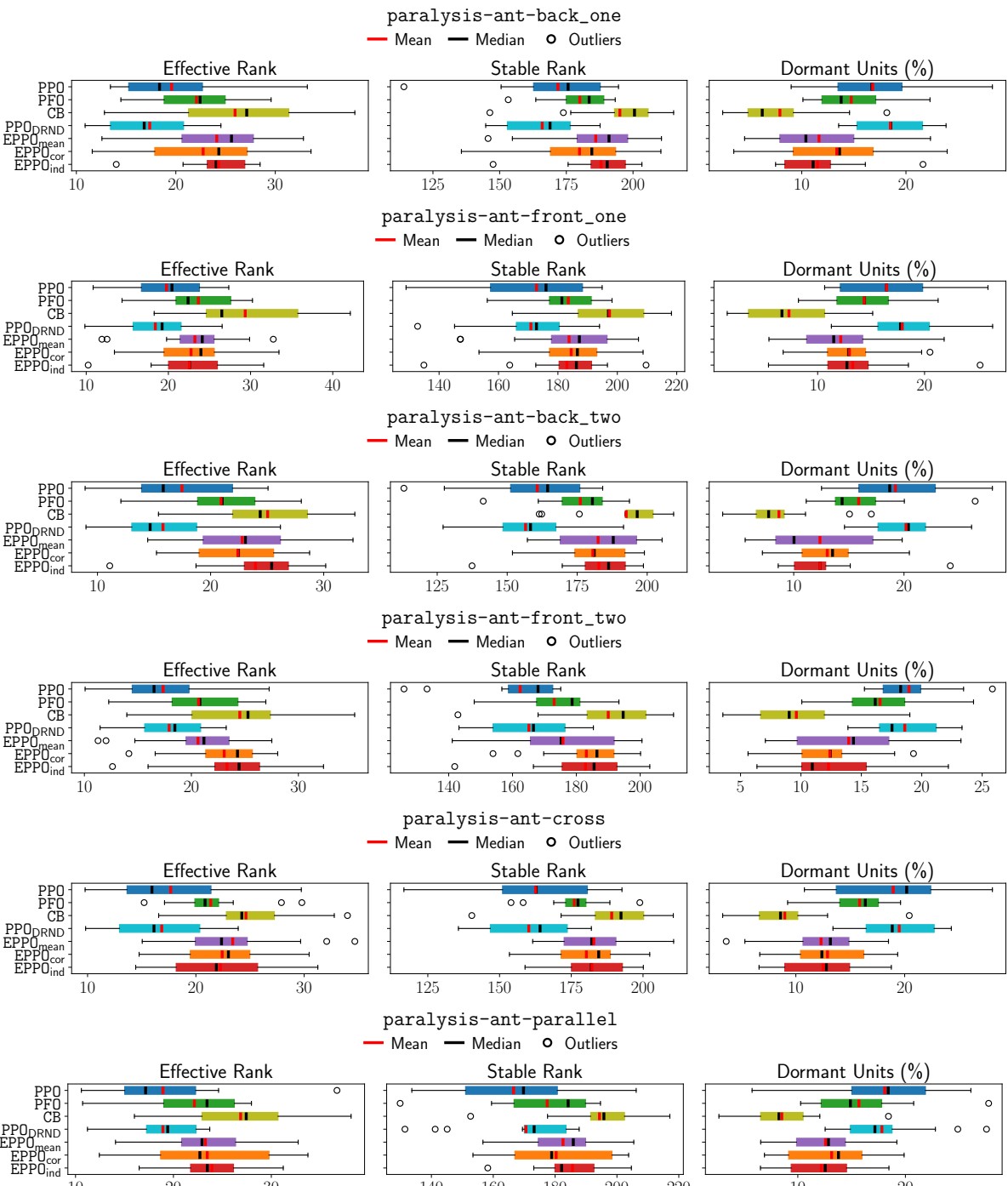

Figure 6: Plasticity preservation analysis for the paralysis experiment on `Ant` environment.

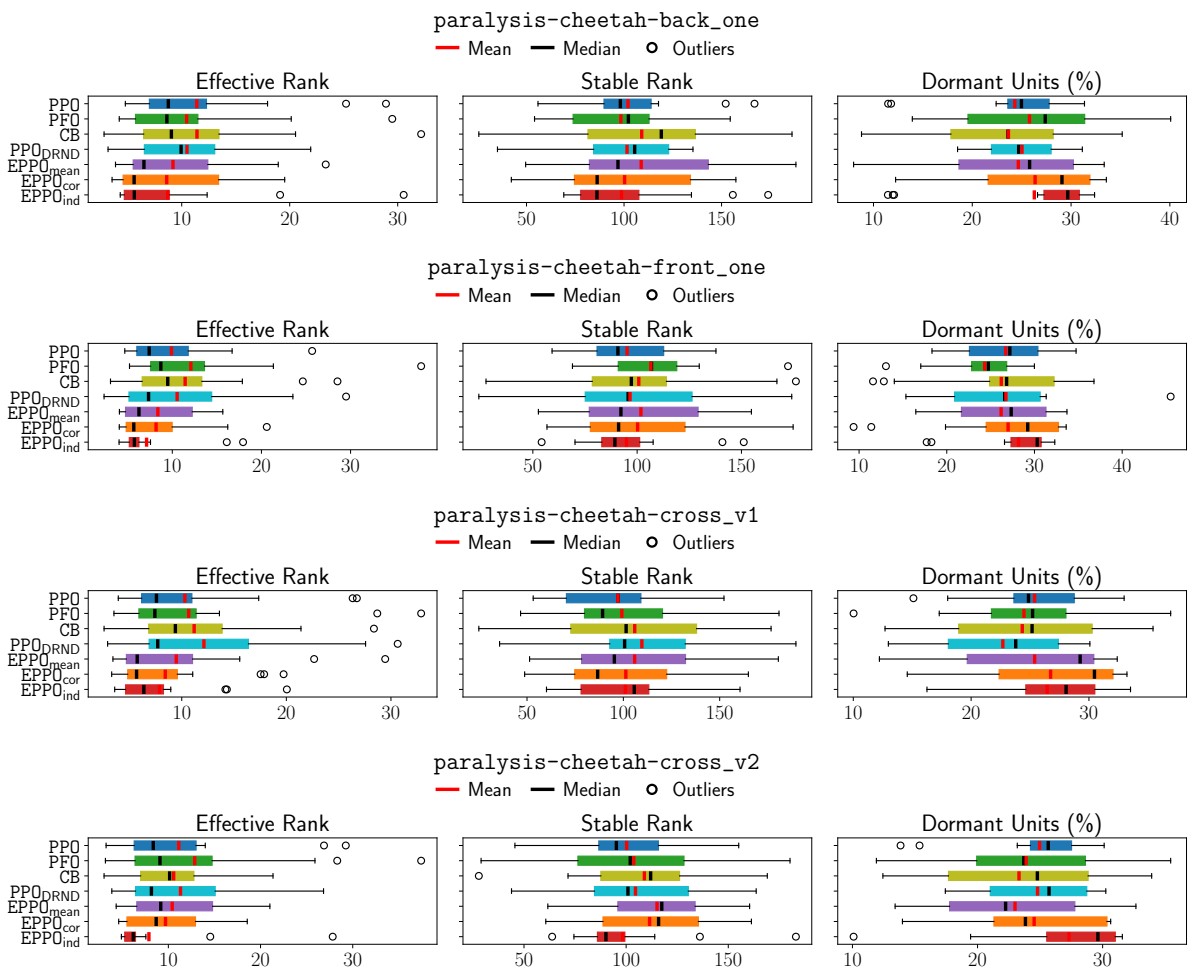

Figure 7: Plasticity preservation analysis for the paralysis experiment on `HalfCheetah` environment.

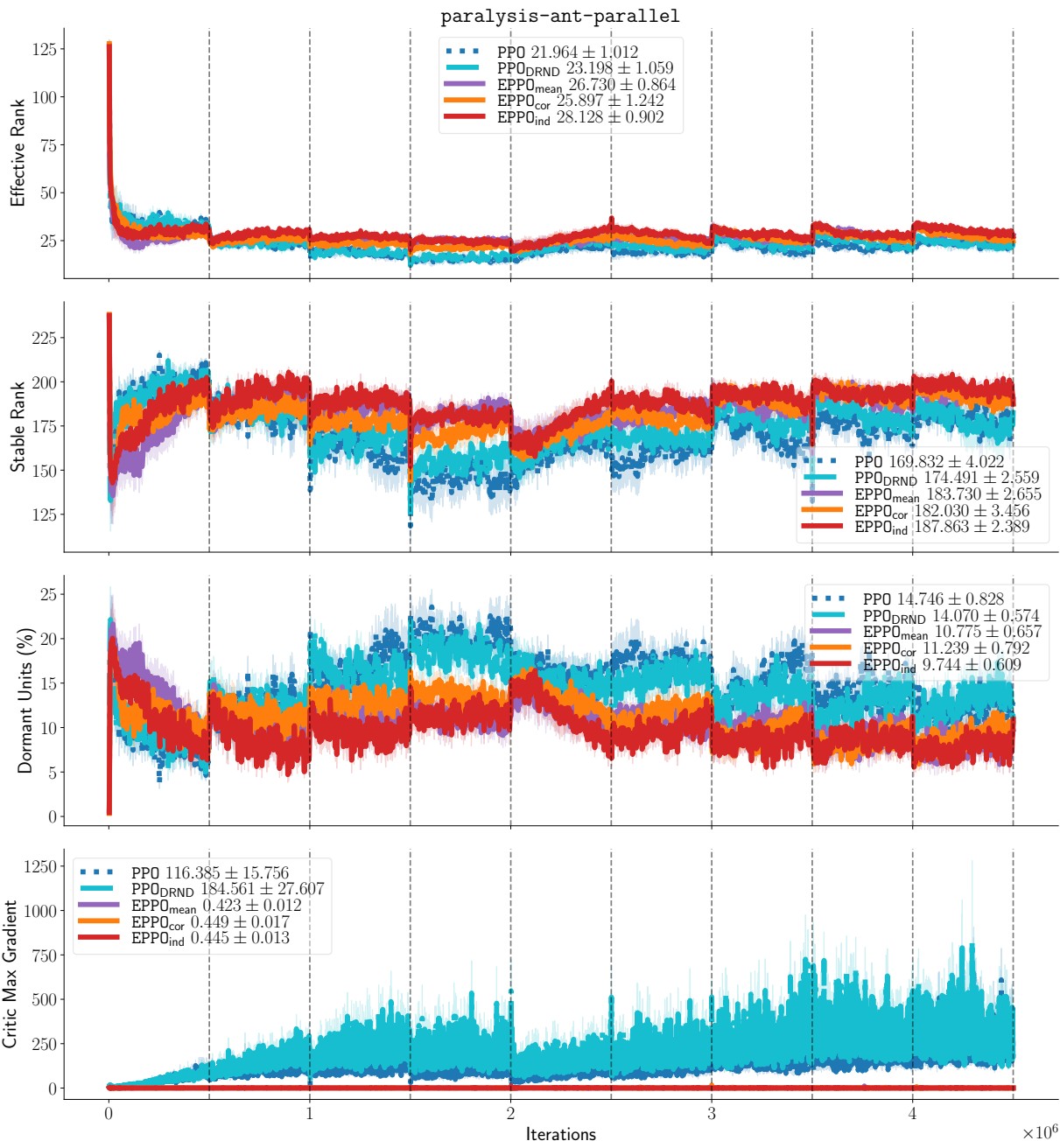

Figure 8: Plasticity preservation metrics and maximum absolute gradient norms throughout training for the paralysis experiment on `Ant-parallel` environment.

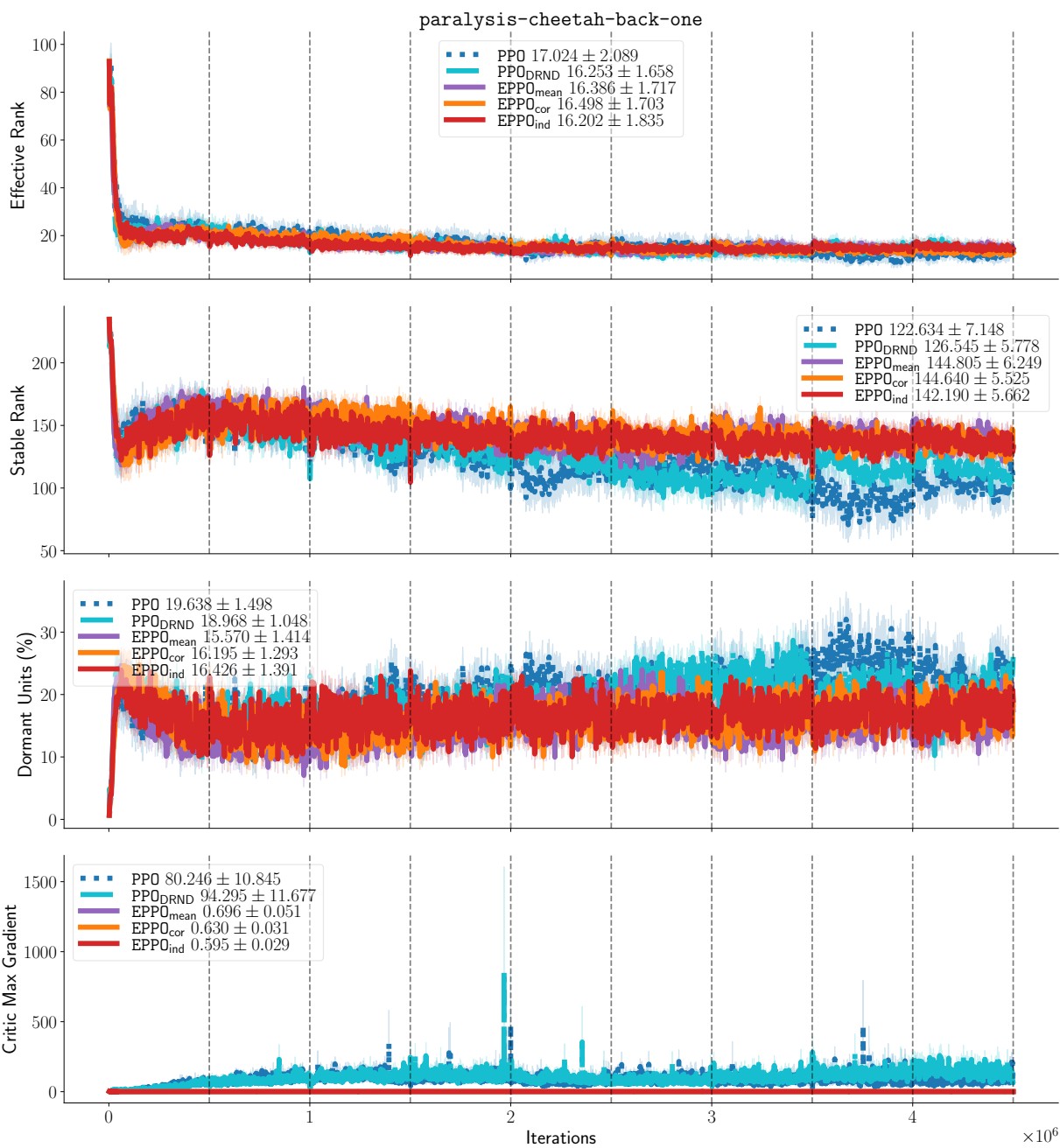

Figure 9: Plasticity preservation metrics and maximum absolute gradient norms throughout training for the paralysis experiment on `HalfCheetah-back-one` environment.

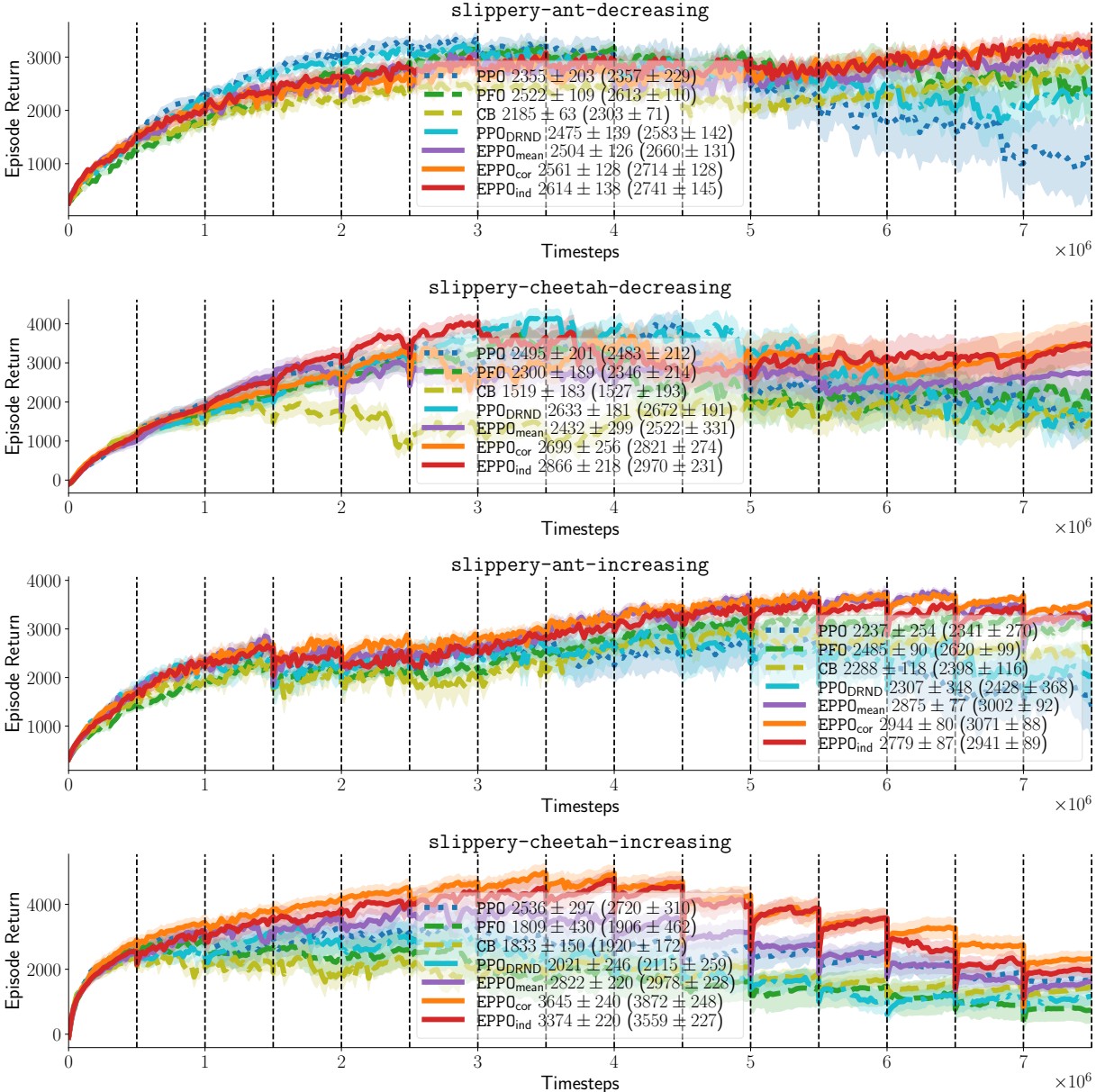

Figure 10: Learning curves for the slippery experiment.

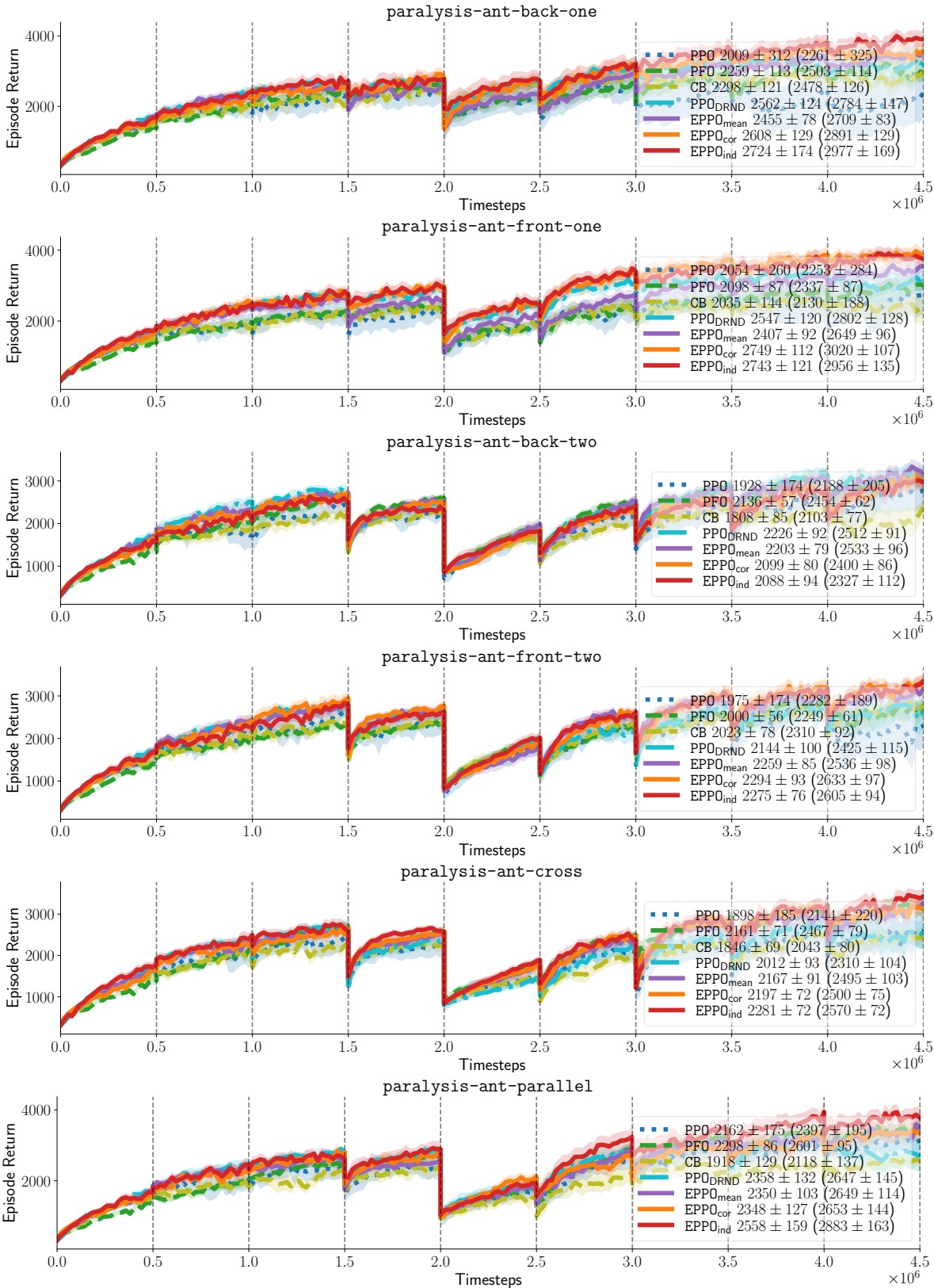

Figure 11: Learning curves for the paralysis experiment on `Ant` environment.

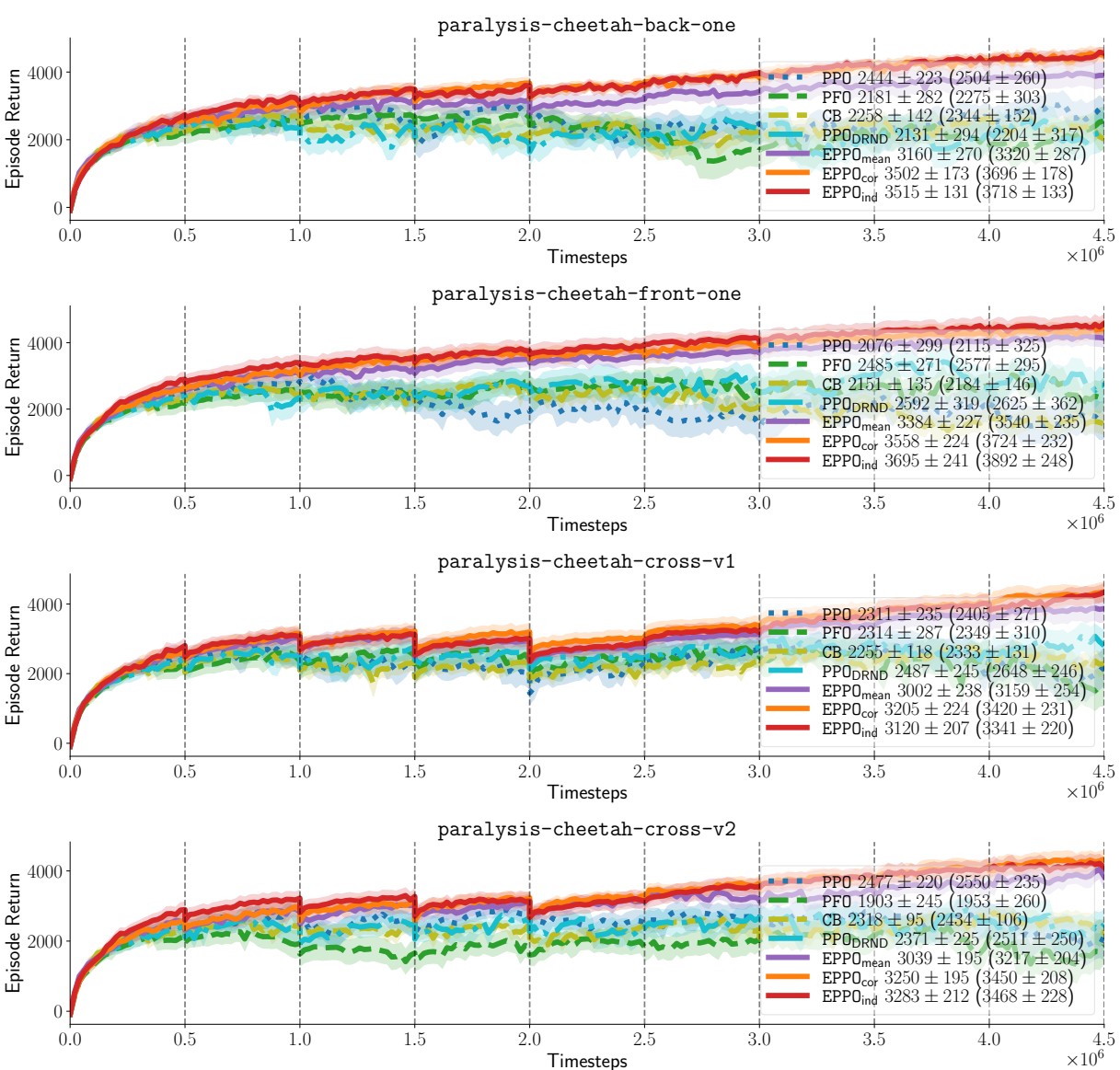

Figure 12: Learning curves for the paralysis experiment on `HalfCheetah` environment.

