# OpenReview forum: "Overcoming Non-stationary Dynamics with Evidential Proximal Policy Optimization"
_TMLR — Accepted by TMLR_

### Review · Reviewer_ToWy · 2025-08-23

**Summary Of Contributions:**

Please summarize the contributions of the paper in your own words. Please also list any key strengths and/or weaknesses, but please be mindful that this is NOT a substitute for the next two text boxes.
 Application of evidential based deep learning to reinforcement learning for uncertainty aware modeling of the value function
Leverage evidential value learning to construct a probabilistic extension of the generalized advantage estimator
Experimental designs tailored to this non-stationary setting

Strengths: paper is well written
Experimental design is well thought out

Weaknesses
I see 3 major problems with this work
1. I am not convinced that this problem is any different than continual learning as described in [1]. The authors have attempted to make a distinction (using discrete tasks), but then use the same environments and problem set-up as [1] which is a continual learning problem. Thus, I am not sure about the framing of the non-stationary problem. If we change the framing of the problem, there are numerous other methods that the authors need to compare their method too. For example, the continual backpropagation algorithm should be compared to the solution of maintaining plasticity & doing better exploration.

2. There is a lack of baselines for comparison. Loss of plasticity, by now, is a well researched problem with many different mitigation strategies. We need to know that the method prescribed in this paper is a useful method compared to other strategies, but we cannot make that decision without other baselines. Similarly, exploration strategies do exist in the literature and we cannot make any decisions about the effectiveness of the proposed method without also comparing to other exploration strategies

3. Statistical testing: I appreciate that the authors showed both box plots & t-tests for the plasticity results but the authors need to include statistical testing of all results, in addition to some clarity on the method that they used to do these tests. For example, I would assume that the data generated by the experiments are not normally distributed and therefore a t-test cannot be used. Did the authors test for normality first? In addition to this, did the authors correct for performing multiple comparisons?

Citations

1. Dohare, S., Hernandez-Garcia, J.F., Lan, Q. et al. Loss of plasticity in deep continual learning. Nature 632, 768–774 (2024). https://doi.org/10.1038/s41586-024-07711-7

**Additional Comments:**

I don’t think [1] proposed dormant unit percentage, I think it was [3].

Citations

3. Ghada Sokar, Rishabh Agarwal, Pablo Samuel Castro, and Utku Evci. 2023. The dormant neuron phenomenon in deep reinforcement learning. In Proceedings of the 40th International Conference on Machine Learning (ICML'23), Vol. 202. JMLR.org, Article 1332, 32145–32168.

**Audience:**

Yes

**Audience Explanation:**

Being able to continually learn/adapt to non-stationarity is an important research direction that the community needs to address. This paper could provide some insights into how to deal with these issues, if certain things are addressed.

**Broader Impact Concerns:**

I don't think this is required.

**Claims And Evidence:**

No

**Claims Explanation:**

1. This problem does not seem to be that different than the problem studied in [1]. The authors of this work use the same environments as [1] but claim that their problem is different. They also claim that continual learning 'assumes a sequential generation of tasks' but this isn't a widely accepted formalism. [1] presents the continual learning problem not as a generation of tasks, but as 'non-stationary' reinforcement learning where the friction of the environment changes which seems to be the exact problem this manuscript attempts to investigate.

2. Loss of plasticity is a known phenomenon and has had several mitigation strategies proposed to overcome it. In a similar vein, there are various exploration strategies that have been proposed to improve exploration. Yet, the submitted work does not compare the proposed method to previous works from either line of work.

3. The statistical results need some clarity on the methods applied & consistency of their application across sets of results. For example the comparisons between algorithms use a ranking, but the plasticity metrics compare using p-values. The authors should explain each of these processes.

**Requested Changes:**

1. Critical: If the authors believe that this is a different problem than that studied in [1], then they need to provide a more thorough rationale for how their problem setting differs from continual learning problems.

2. Critical: building on (1), the authors do not include appropriate baselines for (a) plasticity loss mitigation, or (b) exploration strategies. There is little to no discussion of either of these lines of work in the submitted manuscript, so these areas should be included in the background section.

3. Critical: the authors need to fully explain their methods of statistical testing. This includes statistical testing of all results, in addition to some clarity on the method(s) that they used to do these tests.

4. Critical: It seems that the hyperparameters of included baseline methods were used from previous works, but then the hyperparameters of the proposed method were specifically tuned. The hyperparameters of each method need to be tuned [2] such that fair comparisons can be made between methods.

Citations

2.  Empirical Design in Reinforcement Learning. Andrew Patterson, Samuel Neumann, Martha White, Adam White; 25(318):1−63, 2024.

---

> ### Author Response · Authors · 2025-09-24
> **Response to Reviewer ToWy**
>
> We thank the reviewer once again for the valuable feedback. As noted in our general response above, we have made several revisions that strengthen the paper. Below, we provide point-by-point answers to the reviewer's comments.
>
> ## On the problem of continual vs non-stationarity
> - **Clarified framing:** We have clarified our setting as *non-stationary reinforcement learning* (see the revised paragraph on “Relation of non-stationary reinforcement learning to meta/continual reinforcement learning.”). While related to continual learning, the key distinction lies in the purpose and the related evaluation scheme: Continual learning aims to maximize performance across multiple tasks with retention of past knowledge, whereas non-stationary reinforcement learning focuses on **rapid, on-the-fly adaptation to perpetually changing dynamics under a fixed goal**. Our setup is characterized by structured changes in shorter intervals, which makes rapid adaptation, rather than retention, the central challenge.
>
> - **Added continual backpropagation as baseline & empirical insight:** As per the reviewer’s suggestion, we have added Continual Backpropagation (CB) [1] as a baseline. CB indeed maintains high plasticity but struggles to adapt quickly in our short-interval scenarios, leading to lower returns. This highlights that **plasticity preservation is necessary but not sufficient**; effective performance also requires mechanisms for active discovery and adaptation, which our approach provides through directed exploration.
>
> ## On the choice of baselines and extension of related work
> - **Related work extension:** We expanded the background with discussions on (i) **plasticity mitigation** ("Loss of plasticity mitigation" paragraph) and (ii) **directed exploration** ("Directed exploration" paragraph), providing context for the baselines used in our experiments.
> - **Baselines and experimental coverage:**
> 	- Plasticity preservation: PFO and CB, representing state-of-the-art strategies for plasticity preservation.
> 	- Directed exploration: PPO_DRND, a state-of-the-art method that outperforms PPO_RND [2], which has previously shown success in non-stationary robotics environments [3].
> 	- Total comparison set: Along with plain PPO and three EPPO variants, we evaluate seven models in total. Running these experiments required approximately 22 days on two GPU-powered workstations. We believe these baselines are representative of the most relevant approaches in both plasticity mitigation and directed exploration. This comprehensive set allows us to rigorously test our hypotheses and demonstrate the effectiveness of our proposed method.
>
> ## On the significance test
> Following the reviewer’s suggestion, we applied significance tests to all main results (Tables 1 and 3). For each experiment, we perform a paired one-sided t-test between the best-performing method, as measured by the mean, and all other methods.
> Since this testing procedure is fixed a priori and consists of far fewer comparisons than a fully exploratory scenario (N-1 vs. N*(N-1)/2, where N is the number of methods), we underline all means with a significance level of $p<0.05$ without further correction for multiple testing.
>
> Tables 4–8 in the appendix report the raw p-values for all method pairs, leaving multiple-testing corrections and significance thresholds to the reader. For instance, testing whether any EPPO variant is significantly better than PPO in a given metric and environment shows significance at $p<0.05$ and even $p<0.01$. This result holds under a Bonferroni correction ($p<0.017$ for $p<0.05$ or $p<0.0033$ for $p<0.01$), confirming the robustness of our findings.
>
>
> ## On the hyperparameters
> - **Baseline hyperparameters:** For a fair comparison, all models share the core hyperparameters with plain PPO. For baseline-specific parameters, we followed the recommended values from the original papers. We performed a grid search only on the hyperparameters of CB (details are below) due to reported sensitivity to two hyperparameters. For the rest, no sensitivity has been reported, and we adopted the ones that are most similar to our setup (MuJoCo, online). We report the full set of hyperparameters in Appendix B.3 and Table 10.
> - **CB hyperparameters:** As the original work highlighted sensitivity to the maturity threshold and replacement rate, we conducted a grid search over the ranges suggested in [1], using a representative environment for each type of non-stationarity. Details of the procedure and the final selected values are provided in Appendix B.3 and Table 10. We also observed that CB’s performance is sensitive not only to these parameters but also highly to individual random seeds.
>
> ## On the correction of citation
> We have corrected the reference in the revised manuscript to properly attribute the dormant unit percentage.

---

> > ### Author Response · Authors · 2025-09-24
> > **References**
> >
> > [1] Dohare, et al. *Loss of plasticity in deep continual learning*. Nature (2024).
> >
> > [2] Burda, et al. *Exploration by random network distillation*. ICML (2019).
> >
> > [3] Schwarke, et al. *Curiosity-driven learning of joint locomotion and manipulation tasks*. CoRL (2023).

---

> > > ### Comment · Reviewer_ToWy · 2025-10-07
> > >
> > > I thank the authors for their in-depth replies to both my comments and the comments from my fellow reviewers. An additional thank you for outlining and updating the statistical testing results.
> > >
> > > 1. `non-stationary reinforcement learning focuses on rapid, on-the-fly adaptation to perpetually changing dynamics under a fixed goal. Our setup is characterized by structured changes in shorter intervals` -- I am still failing to see how this is distinct from continual RL. If we take the authors definition of non-stationarity as `structured changes in shorter intervals`, then how does using tasks, with less interaction steps per-task, differ from this? The choice of interaction budget per-task is arbitrary, we could choose a similar interaction budget per-task for any CRL benchmark (i.e. ContinualWorld, COOM) and have an extremely similar problem setting. Thus, I think that the authors have attempted to make a distinction between their (in my view) CRL work and previous CRL works along the wrong axis. When using other problem settings (i.e. ContinualWorld, COOM). One of the issues with these benchmarks, and the ways that they get used in the literature, is the assumptions that they make. For example, that the task ID is given to the agent. Perhaps this may be a better framing for the contributions of EPPO. i.e. `We remove the reliance on certain assumptions from the literature, such as the visibility of the task ID.` I suggest that the authors dive into other CRL works in the literature to highlight the assumptions that are prevalent in the literature.
> > >
> > > 2. I think that the framing of your hypothesis is slightly misaligned with the contributions of the work. The authors make the hypothesis that "the addition of quantifying the uncertainty of the value function enables it to (i) preserve plasticity and (ii) explore effectively." But the uncertainty quantification is not a plasticity preservation technique, it is an exploration technique.
> > > Thus, how does uncertainty quantification & exploration maintain network plasticity? I propose the authors perform the following experiment to make a first attempt to try and answer this question. Throughout the course of training, once the agent has filled the PPO/EPPO agent's buffer, attempt to quantify the diversity of the data in the buffer. This diversity of data through better exploration seems like a relevant piece of the puzzle to maintaining plasticity naturally, and the threads of this idea can be found in some recent works such as [1][2]. [1] leverage UMAP to quantify their state coverage, outlined in section B.7.
> > >
> > > 1.The Impact of On-Policy Parallelized Data Collection on Deep Reinforcement Learning Networks, Walter Mayor, Johan Obando-Ceron, Aaron Courville, Pablo Samuel Castro, ICML 2025.
> > >
> > >
> > > 2. Multi-Task Reinforcement Learning Enables Parameter Scaling, Reginald McLean, Evangelos Chatzaroulas, J.K. Terry, Isaac Woungang, Nariman Farsad, Pablo Samuel Castro, RLC 2025.

---

> > > > ### Author Response · Authors · 2025-10-08
> > > >
> > > > Thank you for your follow-up questions.
> > > >
> > > > ## On the distinction between continual RL and our setup
> > > >
> > > > As the reviewer notes, when the environment is structured as discrete tasks with shorter interaction budgets, it could also be interpreted as a continual RL setup with finer time granularity. However, our objective fundamentally differs: continual RL emphasizes **knowledge retention** and **avoiding catastrophic forgetting** across a task sequence, while our focus is on **rapid, on-the-fly adaptation** to ongoing changes [1].
> > > >
> > > > To clarify our framing, we highlight several assumption differences between our setup and the cited standard continual RL benchmarks [2, 3] as the reviewer suggests: (i) we do not assume access to task IDs, (ii) we do not rely on task-specific replay buffers, and (iii) we do not train task-specific network heads. We will incorporate this distinction into the paper once the reviewer agrees with this formulation.
> > > >
> > > >
> > > > ## Relation of uncertainty and plasticity
> > > > > But the uncertainty quantification is not a plasticity preservation technique, it is an exploration technique.
> > > >
> > > > Uncertainty quantification in this work is not merely an exploration technique. EPPO leverages evidential uncertainty in two complementary ways: it shapes how the critic learns by maintaining a value distribution, and it shapes how the actor explores by using the inferred distribution to construct an upper confidence bound on the advantage. Given the hyperprior $p(\mathbf{m})$, the evidential loss penalizes overconfidence through its regularizing effect, which requires balancing the data fit with the prior assumption, as in a Bayesian framework. This increases critic uncertainty under distributional shift, which, in turn, reactivates gradient flow and helps the critic remain flexible and adaptive (Section 1). As shown in Figure 3, even EPPO_mean, which omits any exploration bonus ($\kappa=0$), already improves upon PPO and PPO_DRND in all plasticity metrics. Adding an exploration bonus (EPPO_cor and EPPO_ind) or an exploration mechanism alone (PPO_DRND) does not further enhance plasticity but, as evidenced in Tables 1-2, substantially improves overall task performance through better-directed exploration. Thus, evidential uncertainty simultaneously supports critic plasticity and enables more effective exploration.
> > > >
> > > > The recent works [4, 5] provide interesting insights into the relationship between data diversity and plasticity. To examine this in the context of EPPO, we perform an analysis inspired by [4], quantifying the diversity of experiences throughout training. Our results show that EPPO variants outperform PPO in the state-space coverage metric introduced in [4]. This increased coverage may indeed complement EPPO’s preserved plasticity, consistent with [4].
> > > >
> > > >
> > > > **Experiment details:**
> > > > We selected the _paralysis_ and _cheetah_ environments with the _back-one_ strategy. These were chosen because they offer the shortest computation time given their relatively low number of tasks and state dimensionality. The _back-one_ strategy was chosen arbitrarily. We trained both plain PPO and EPPO variants on this setup using a single seed (seed = 1) due to limited available time and saved the states used for policy updates. Following [4], we applied UMAP with default parameters using the implementation from [6] to project the states into a 2D space, which is necessary due to the large number of states. We then applied min–max normalization with a small epsilon ($1e−5$) to scale the projected data to the range $[0, 1)$. The remaining steps follow the coverage calculation described in Appendix B.7 of [4]. We used a grid size of $G = 100$, corresponding to a $0.01$ resolution per grid cell. As [4] does not report the specific choices of UMAP parameters, scaling method, or grid size, we selected these values to ensure fairness and reproducibility.
> > > >
> > > > |                            |   PPO  | EPPO_mean | EPPO_cor | EPPO_ind |
> > > > |:--------------------------:|:------:|:---------:|:--------:|:--------:|
> > > > | paralyzed cheetah back one | 0.0097 |   0.1178  |  0.1326  |  0.1151  |
> > > >
> > > > **References**
> > > >
> > > > [1] Khetarpal, et al. *Towards continual reinforcement learning: a review and perspectives*. JMLR (2022).
> > > >
> > > > [2] Wołczyk, et al. *Continual world: a robotic benchmark for continual reinforcement learning*. NeurIPS (2021).
> > > >
> > > > [3] Tomilin, et al. *A game benchmark for continual reinforcement learning*. NeurIPS (2023).
> > > >
> > > > [4] Mayor, et al. *The impact of on-policy parallelized data collection on deep reinforcement learning networks*. ICML (2025).
> > > >
> > > > [5] McLean, et al. *Multi-task reinforcement learning enables parameter scaling*. RLC (2025).
> > > >
> > > > [6] https://docs.rapids.ai/api/cuml/stable/api/#umap

---

> > > > > ### Comment · Reviewer_ToWy · 2025-10-08
> > > > >
> > > > > Thank you for the reply.
> > > > >
> > > > > In Section 2.2 of [1] it is also noted that for CRL agents "... it is not immediately clear if agents must perform well on all previously seen tasks," while also stating that "... quick adaptation and building on relevant previously learned behaviors are also central to the study of continual RL." I only continue to bring this up because of the rigidity of the author's claims that their work is strictly not CRL, while there isn't a single definition of CRL that the field has some sort of consensus on. In my view this work is in fact a CRL problem but many of the common assumptions (as noted in the previous comment) are weakened. In fact, the problem that the authors attempt to solve here with these weakened assumptions, is much more difficult than the problems solved in previous works [2, 3] that leverage those stronger assumptions about what is available to the agent or how the agent learns.
> > > > >
> > > > > I thank the authors for these preliminary experiments into state coverage. While computationally expensive, I do ask that the authors perform more of these experiments across additional random seeds in order to gain additional insights into the role that state coverage/exploration plays here. As hinted to in my previous comment, I think that this thread that [4], [5], and the authors submission starts to pull at is important: maintaining network plasticity without an explicit mechanism for doing so.
> > > > >
> > > > > As it stands, the EPPO method remains somewhat of a black box. While I understand the Bayesian formulation and the authors' explanation that 'the evidential loss penalizes overconfidence through its regularizing effect, which requires balancing the data fit with the prior assumption,' the manuscript currently demonstrates only that an evidential value function improves performance metrics. I encourage the authors to investigate the mechanisms underlying this improvement.
> > > > > Specific questions that would strengthen the contribution: Does the evidential framework (with or without exploration) behave in a similar manner to [4] and [5] in increasing the amount of data that is used?  Does the evidential framework make optimization easier or more stable? If so, through what mechanism: does it improve gradient flow, reduce interference between updates, or something else? How do alternative value function training methods (evidential vs. regression vs. classification) affect the learned representations? What is the causal pathway from uncertainty quantification to plasticity preservation? Does it prevent specific pathologies like dormant neurons or rank collapse? Can the authors provide diagnostic metrics (e.g., effective rank, dead neuron counts, representation similarity) throughout training?
> > > > >
> > > > > Addressing these questions would elevate the work from an empirical observation to mechanistic insight about why uncertainty-aware value learning maintains plasticity under non-stationarity. I think that the mentioned state-coverage idea could be a reasonable place to start. I kindly ask that the authors attempt to answer these questions **empirically** rather than anecdotally.
> > > > >
> > > > > [1] Khetarpal, et al. Towards continual reinforcement learning: a review and perspectives. JMLR (2022).
> > > > >
> > > > > [2] Wołczyk, et al. Continual world: a robotic benchmark for continual reinforcement learning. NeurIPS (2021).
> > > > >
> > > > > [3] Tomilin, et al. A game benchmark for continual reinforcement learning. NeurIPS (2023).
> > > > >
> > > > > [4] Mayor, et al. The impact of on-policy parallelized data collection on deep reinforcement learning networks. ICML (2025).
> > > > >
> > > > > [5] McLean, et al. Multi-task reinforcement learning enables parameter scaling. RLC (2025).

---

> > > > > > ### Author Response · Authors · 2025-10-17
> > > > > >
> > > > > > Thank you for your continued interaction.
> > > > > >
> > > > > > ## On the relation of our setup and continual RL
> > > > > >
> > > > > > Our emphasis on distinguishing our approach from continual learning stems from assuming a weaker signal and focusing solely on adaptation, rather than knowledge/ability preservation (e.g., [1] discusses the necessity of "resistance to catastrophic forgetting" just before the quoted sentence). However, we agree that, as our agent must continually adapt to a changing, non-stationary environment, it can be understood as a form of continual learning. We revised the _Relation of non-stationary reinforcement learning to curriculum, meta-, and continual reinforcement learning_ paragraph in Section 2.1 considering reviewer's suggestions.
> > > > > >
> > > > > > ## On the coverage experiments
> > > > > >
> > > > > > As the reviewer suggested, we conducted additional coverage experiments for PPO, PPO_DRND, and all EPPO variants (see Table 5, _Exploration analysis_ paragraph in Section 4.1, and Appendix B.1.4).
> > > > > >
> > > > > > > Does the evidential framework (with or without exploration) behave in a similar manner to [2] and [3] in increasing the amount of data that is used?
> > > > > >
> > > > > > Effective adaptation in non-stationary environments requires agents to explore a wide range of states, particularly high-reward regions, while preserving plasticity to continually adjust to newly visited states. EPPO achieves this by combining directed exploration with plasticity preservation within a single probabilistic framework (see the _Plasticity preservation through evidential value learning_ paragraph in Section 4.1). Unlike approaches that rely on multiple environments [2] or multiple tasks [3], evidential value learning naturally encourages diverse data collection. Our results show that the directed exploration component increases data diversity (see Table 5 for PPO_DRND and EPPO results). However, we cannot conclude that data diversity alone drives plasticity: although PPO_DRND collects more diverse data than plain PPO, its plasticity scores remain comparable (see the PPO vs. PPO_DRND comparison in Table 5 and Figure 3).
> > > > > >
> > > > > > ## On further empirical results
> > > > > >
> > > > > > > Does the evidential framework make optimization easier or more stable? If so, through what mechanism: does it improve gradient flow, reduce interference between updates, or something else?
> > > > > >
> > > > > > Yes, it makes optimization more stable by smoothing and regularizing the loss landscape, which improves gradient flow and prevents dormant units (see the _Plasticity preservation through evidential value learning_ paragraph in Section 4.1).
> > > > > >
> > > > > > > How do alternative value function training methods (evidential vs. regression vs. classification) affect the learned representations?
> > > > > >
> > > > > > Regarding evidential regression versus regular regression, PPO minimizes a squared error to learn the value function, i.e., performs regular regression. See Algorithm 1 in the appendix of the paper, which summarizes each loss term. A switch from regression to classification, as suggested, e.g., by [4], would make the evidential counterpart the original Dirichlet prior-based method by [5]. We expect any findings on the relationship between evidential regression vs plain regression to translate to evidential classification vs plain classification without providing additional insight. Therefore, such a comparison is outside the scope of this work.
> > > > > >
> > > > > > > What is the causal pathway from uncertainty quantification to plasticity preservation?
> > > > > >
> > > > > > Evidential value learning provides uncertainty quantification through its evidential parameters. This uncertainty allows the agent to perform directed exploration, guiding it toward more diverse and high-reward states. The evidential parameters make the loss function of the value learning stage more amenable to gradient-based training, thereby improving the plasticity of the critic network. See the _Plasticity preservation through evidential value learning_ paragraph in Section 4.1 for the details.
> > > > > >
> > > > > >
> > > > > > > Does it prevent specific pathologies like dormant neurons or rank collapse?
> > > > > >
> > > > > > Yes, it prevents dormant units and excessive gradient spikes by dynamically scaling gradient updates. See Figures 8 and 9, which show that EPPO variants maintain lower critic gradient norms and fewer dormant units throughout training compared to mean-squared-error baselines (PPO and PPO_DRND).
> > > > > >
> > > > > > > Can the authors provide diagnostic metrics (e.g., effective rank, dead neuron counts, representation similarity) throughout training?
> > > > > >
> > > > > > We now provide diagnostic metrics throughout training, including effective rank, stable rank, dormant units, and the critic's maximum gradient (see Figures 8 and 9).

---

> > > > > > > ### Author Response · Authors · 2025-10-17
> > > > > > > **References**
> > > > > > >
> > > > > > > [1] Khetarpal, et al. *Towards continual reinforcement learning: a review and perspectives*. JMLR (2022).
> > > > > > >
> > > > > > > [2] Mayor, et al. *The impact of on-policy parallelized data collection on deep reinforcement learning networks*. ICML (2025).
> > > > > > >
> > > > > > > [3] McLean, et al. *Multi-task reinforcement learning enables parameter scaling*. RLC (2025).
> > > > > > >
> > > > > > > [4] Farebrother et al. *Stop regressing: training value functions via classification for scalable deep RL*, ICML (2024)
> > > > > > >
> > > > > > > [5] Sensoy, et al. *Evidential deep learning to quantify classification uncertainty*. NeurIPS (2018).

---

> > > > > > > > ### Comment · Reviewer_ToWy · 2025-10-17
> > > > > > > >
> > > > > > > > Thank you for your hard work in answering these questions. This analysis makes the work very strong. I think some commentary on the connections between [2], [3], and the submitted work is needed in the "Loss of plasticity mitigation" background section. Other than that, I am very happy with the extra work that the authors have put into their work.

---

> > > > > > > > > ### Author Response · Authors · 2025-10-17
> > > > > > > > >
> > > > > > > > > We thank the reviewer for helping us improve the paper. We learned a lot from the valuable feedback and will add the comments discussed above regarding [2] and [3] to the _Loss of plasticity mitigation_ paragraph in the _Background_ section as requested.

---

### Review · Reviewer_iEJd · 2025-08-29

**Summary Of Contributions:**

The paper studies the design of deep reinforcement learning algorithms under environment nonstationarity. The authors propose Evidential Proximal Policy Optimization (EPPO), an algorithm built on top of PPO that integrates evidential deep learning into the critic. They provide the option of implementing the algorithm in combination with directed exploration, a technique for increasing the exploration of high-uncertainty state and action regions. The authors conduct a range of experiments in (modified) MuJoCo environments and show that empirically the proposed algorithm achieves higher scores measured by area under learning curve and final return.

On the positive side, the paper clearly motivates, introduces, and evaluates the proposed method. It is generally well written --  the existing works & their shortcomings, the technical development of this work, environment setup, discussion of experimental results are mostly clear. The experiments, though only conducted on two environments, are sufficiently comprehensive and consistently demonstrate the better performance of the proposed algorithm over the baselines.

At the same time, I find the paper lacking in several aspects.

1) The proposed algorithm is a hybrid of PPO, evidential learning, and directed exploration, making the paper an "A+B" type of work with limited technical contribution. While works of this nature can be important/impactful/interesting to the community, I personally believe that the bar for such combinations should be higher and would hope to see more depth in terms of experiments in larger-scale problems, mathematical analysis, and/or more thorough ablation studies (especially regarding the claim on the importance of plasticity).

2) While the presentation is mostly clear, I find Sections 2.2 and 3.1 hard to follow, having no background knowledge on evidential deep learning. I particularly have problems understanding what innovations are needed to make evidential learning applicable to the estimation of value functions, and would appreciate the authors' clarification.

3) While the performance from the proposed method is consistently better, the results are often not statistically significant, with the differences between EPPO variants and baselines falling within overlapping error bars.

**Audience:**

Yes

**Audience Explanation:**

While evidential learning and directed exploration have been studied, their integration with PPO and application in the context of nonstationary RL is new. It is good to see that these techniques lead to better solution of nonstationary RL problems and I believe researchers in the community will find this work interesting.

**Broader Impact Concerns:**

No concerns.

**Claims And Evidence:**

Yes

**Claims Explanation:**

The authors posit that two properties of an algorithm play important roles in dealing with nonstationarity. The first is plasticity of the critic and the second is directed exploration. This motivated the algorithm proposed in the work, and the contribution of the two properties to algorithm performance in nonstationary RL environments are comprehensively studied through experiments.

**Requested Changes:**

I have the following questions/comments that I hope the authors can clarify, besides those raised in the first text box.

1) The formulation introduced in Section 2.1 considers a time-homogeneous reward function and a time-dependent transition kernel. Usually such assumptions on reward/transition kernel fixed over time are made for the purpose of mathematical analysis. However, as this paper does not contain theoretical results, I wonder why the authors need to impose the time homogeneity of reward?

2) Some references seem cited in the introduction section with no clear purpose. For example, in "On-policy algorithms, such as Proximal Policy Optimization (PPO) (Schulman et al., 2017), are particularly well-suited for non-stationary environments (Sutton et al., 2007) because they rely solely on data from the most recent policy, ensuring policy improvement through sufficiently small updates (Kakade & Langford, 2002)." --  how does the work of Sutton et al. and Kakade & Langford relate to the discussion here?

3) I do not think the experimental results conclusively show that higher critic plasticity leads to better performance in nonstationary environments. The benefit from directed exploration is more convincing, as the authors can carry out experiments with and without this component and compare. However, plasticity is not directly controllable. Across the different algorithms (and variants), plasticity is different, but so are many other factors. This makes it inaccurate to attribute the observed performance differences specifically to plasticity. At best we can only say there is some correlation. Furthermore, the separation in effective rank, stable rank, and dormant units across is not always large enough and well within the standard error range.

---

> ### Author Response · Authors · 2025-09-24
> **Response to Reviewer iEJd Part 1/2**
>
> We thank the reviewer once again for the valuable feedback. As noted in our general response above, we have made several revisions that strengthen the paper. Below, we provide point-by-point answers to the reviewer's comments.
>
> ## On the perceived ‘A+B’ nature and experimental scope
> - **Beyond ‘A+B’.** Our contribution introduces novel elements:
> 	- We connect two previously disjoint directions&mdash;evidential learning and non-stationary reinforcement learning&mdash;within a principled RL framework. This integration is non-trivial because evidential learning is designed for static conditions, whereas non-stationary RL demands continual adaptation to changing dynamics. Bridging these perspectives requires ensuring that uncertainty estimates remain meaningful when the environment dynamics are changing. Our approach uses uncertainty to guide exploration, enabling the agent to detect and respond to distributional shifts more effectively.
> 	- We improve deep evidential regression [1] by introducing Bayesian hyperpriors, addressing limitations of prior heuristics [1,2].
> - **Experimental scale.** Our setting features non‑stationary environments with multiple variants and repetitions, which is the regime where plasticity and exploration matter most. We evaluate four strong baselines (including a new one), and the study required approximately 22 GPU‑days across two workstations (see "Compute Time" paragraph of Experiments section).
> - **Mathematical analysis.** We acknowledge the lack of formal guarantees. Even without the added complexity of the evidential pipeline, providing formal guarantees when combining non-stationarity, non-tabular state-action spaces, and non-linear value functions is a highly nontrivial task for which, to the best of our knowledge, no results are known in the literature.
> This limitation is common to our method and the baselines. We've added this limitation to Section 5.
> - **Ablations & plasticity.** The experiments constitute a comprehensive ablation:
> 	1. Plasticity preservation: PFO, CB, EPPO_mean
> 	2. Directed exploration: PPO_DRND, EPPO_ind, EPPO_cor
> 	3. Baseline: plain PPO
>
> Results consistently show that combining plasticity preservation with directed exploration outperforms using either of these components alone.
>
> ## On the presentation of Evidential Deep Learning
> We have expanded Sections 2.2 and 3.1 with more background and references to make evidential deep learning (EDL) accessible to readers without prior knowledge. A brief summary is as follows:
> - **EDL for classification:** Originally introduced by [3], EDL predicts not only a point estimate (e.g., the probability of class c) but also a distribution over categorical probabilities via a Dirichlet distribution. This allows the model to distinguish between high-evidence (confident) and low-evidence (uncertain) predictions.
> - **EDL for regression:** [1] extended EDL to regression tasks, modeling outputs as a normal distribution with mean and variance parameters governed by a Normal-Inverse-Gamma prior. This provides principled uncertainty estimates over continuous values.
> - **Adapting EDL to value functions:**
> The assumption that value function estimates follow a normal likelihood is a modeling choice motivated in evidential regression by the fact that it allows for a closed-form, analytical objective, i.e., the marginal likelihood given by the Student-t distribution. While other combinations are possible in principle, these would lead to a less stable, sampling-based objective.
> - **Challenges and our contribution:** Prior works [2, 4] observed strong instability in regression setups, which we also encountered in value learning. Unlike prior heuristic fixes, we introduce a principled, stability-inducing technique motivated by Bayesian hyperpriors, which improves training stability and reliability in value estimation.
>
> For a more comprehensive introduction to EDL, see [5].

---

> > ### Author Response · Authors · 2025-09-24
> > **Response to Reviewer iEJd Part 2/2**
> >
> > ## On the performance and overlapping error bars
> >
> > - **Main tables:** In slippery environments (Table 1) and paralysis environments (Table 3), EPPO outperforms all baselines by at least one standard error in nearly all cases (15 seeds per setting). We consider this strong evidence in favor of EPPO compared to our PPO-based baselines.
> > - **On overlapping box plots:**  These plots aggregate across seeds, environments, and strategies, which inflates overlap. Tables 4–9 in the appendix show that, at the individual scenario level, differences between EPPO variants and PPO are statistically significant for plasticity preservation metrics.
> >
> > ## On the time-homogeneous reward function and time-dependent transition kernel
> > We revised the text to clarify this point. The reward function is time-homogeneous because all tasks share the same overall goal, while the environment dynamics change over time. Non-stationarity arises from these time-dependent transitions, not from changes in the reward function. This formulation allows us to isolate adaptation to changing dynamics while keeping the task objective fixed.
> >
> >
> > ## On the references
> > Thank you for catching the error regarding the Sutton et al. reference. We have removed it and revised the introduction so that each citation directly supports the statement it accompanies. Kakade & Langford (2002), however, is relevant, as they are the seminal conservative policy iteration paper deriving optimal policy update step sizes.
> >
> > ## On the plasticity
> > Our results show that combining plasticity preservation with directed exploration consistently improves performance in non-stationary environments, whereas using only one of these components yields suboptimal outcomes. Importantly, the new baseline we introduced (continual backpropagation (CB) [6]) demonstrates only higher plasticity but is clearly behind in performance, which supports our hypothesis that plasticity is beneficial only when combined with directed exploration.

---

> > > ### Author Response · Authors · 2025-09-24
> > > **References**
> > >
> > > [1] Amini, et al. *Deep evidential regression*. NeurIPS (2020).
> > >
> > > [2] Meinert, et al. *The unreasonable effectiveness of deep evidential regression*. AAAI (2023).
> > >
> > > [3] Sensoy, et al. *Evidential deep learning to quantify classification uncertainty*. NeurIPS (2018).
> > >
> > > [4] Oh, et al. *Improving evidential deep learning via multi-task learning*. AAAI (2022).
> > >
> > > [5] Ulmer, et al. *A survey on evidential deep learning methods for uncertainty estimation*. TMLR (2023).
> > >
> > > [6] Dohare, et al. *Loss of plasticity in deep continual learning*. Nature (2024).

---

> > > > ### Comment · Reviewer_iEJd · 2025-09-30
> > > >
> > > > I appreciate the authors' clarification. I think the first bullet in the response makes sense -- while the work builds on PPO, evidential learning, and directed exploration, how they can be combined for RL in nonstationary environment is previously unclear, and integrating the pieces together to make the system work is a non-trivial effort.
> > > >
> > > > My biggest unaddressed concern is on the claimed benefit of critic network plasticity. From the original paper I got the sense that the authors argue higher plasticity is good independent of directed exploration. However, in light of the response and the new baseline performance, the authors suggest that higher plasticity is only beneficial in the presence of directed exploration. I would like to if I misunderstand anything on this point, and I suggest the authors revising the language in the paper to avoid causing confusion.
> > > >
> > > > I raised an issue on whether we can decisively conclude plasticity is beneficial from the current experimental results, and I believe it remains unaddressed. "Plasticity is not directly controllable. Across the different algorithms (and variants), plasticity is different, but so are many other factors. This makes it inaccurate to attribute the observed performance differences specifically to plasticity. At best we can only say there is some correlation."

---

> > > > > ### Author Response · Authors · 2025-09-30
> > > > >
> > > > > Thank you for your follow-up question.
> > > > >
> > > > > ## On the claims about plasticity
> > > > > Already in the original submission, our main hypothesis was that plasticity preservation and directed exploration need to be present simultaneously (see the second paragraph in the introduction). In fact, our claim is that plasticity is beneficial only when combined with directed exploration. In light of the new baseline results, we revised the discussion (see the Discussion paragraph in Section 4.1) to (i) explicitly compare the plasticity of EPPO variants with plain PPO, showing clear improvements, and (ii) add a bullet point stating that neither plasticity preservation nor directed exploration alone is sufficient to handle non-stationary environments. We believe the revised text now reflects this point, but we are happy to revise the manuscript further if our answer does not fully clarify the concerns.
> > > > >
> > > > > ## On the plasticity
> > > > > Our experiments are designed to separate the contributions of the two ingredients in our hypothesis. PPO_DRND introduces directed exploration but does not preserve plasticity; it outperforms plain PPO in adaptation (see Tables 1 and 3) while having almost identical plasticity as plain PPO (see Figure 3 and the individual results in Figures 5 to 7). However, its directed exploration mechanism is not sufficient to achieve the best performance. Conversely, CB emphasizes plasticity preservation (and scores highest on plasticity metrics) but struggles to adapt quickly to environmental changes. The best-performing models, the EPPO variants with directed exploration, combine both properties. This suggests that neither plasticity preservation nor directed exploration alone suffices to handle non-stationarity, but their combination is effective.
> > > > >
> > > > > That said, we agree that attributing performance differences solely to plasticity is difficult because plasticity is not directly controllable and may correlate with other factors. Indeed, CB provides a counter-example where high plasticity does not translate into the best adaptation performance.

---

> > > > > > ### Comment · Reviewer_iEJd · 2025-10-03
> > > > > >
> > > > > > I thank the authors for the further remarks. I think of the work positively and will recommend acceptance. However, I personally think that the design, implementation, and experimentation of the proposed algorithm already consist good contribution, and it would be better not to discuss the hypothesis that the algorithm is better because of the combination of exploration and critic plasticity, which does not seem sufficiently supported by the current experiments.

---

> > > > > > > ### Author Response · Authors · 2025-10-03
> > > > > > >
> > > > > > > We thank the reviewer for the constructive feedback and positive recommendation.

---

### Review · Reviewer_5uv8 · 2025-09-11

**Summary Of Contributions:**

The paper introduces a distributional formulation to the GAE module in PPO, which is parameterized explicitly with some Gaussian family and uses the distribution-aware objective to compute a mean-value risk measure to guide the policy. The idea is interesting, and the problem set is relevant in robotics. But I have a few serious concerns:
1. The authors claim "Rapid adaptation to each new situation is desired instead of remembering all previous situations." From the description of this work, learning in non-stationary dynamics seems to be the same as continual learning, which also adapts to new environments. I am not sure why this work necessarily needs to NOT remember all previous situations, which I believe is even more difficult. Isn't remembering all scenarios the ultimate solution for either setup?

2. The formulation in (3) is precisely the distribution RL with risk measure being the mean-variance risk measure. To this end, the authors need to discuss risk-sensitive control. Furthermore, in this case, the usual Dist RL parameterization has much more freedom (e.g., approximating the distribution function). What is the advantage of this method against dist RL using, say, directly parameterize distribution functions? I am sure one can also regularize with some prior as well.

3. Looking at the results in the paper and appendix, I think at the beginning stage PPO is already not as good as the proposed method, which suggests that the distributional formulation has additional effect, in which I require the authors to compare with distributional RL and risk-sensitive control methods as well.

4. The setup of the experiment and the evaluation metric seems too restricted and results are not very convincing. There are a multitude of robotics work (in locomotion, which is the same task as in this work) that deal with either slippery surface (friction coefficient down to 0.1) or even rough terrain which can all be solved by PPO with state-of-the-art performance. With curriculum training, the tasks are essentially the same as in the experiment section. So I am perplexed by the presented result, which does not make sense to me for both the AULC and final return metric. It seems to me that this might be some implementation issue. Again, it is natural that a sophisticated design can bring higher performance. Then the question is, what benefits does the proposed method really have if other method with more slightly different technique can solve the same problem (if not even better)?

**Audience:**

Yes

**Audience Explanation:**

This paper has a few elements that are interesting, e.g., regarding changing environment dynamics, using distributional critic and risk sensitive control. The application field in robotics is also highly relevant these days.

**Broader Impact Concerns:**

No ethical concerns as far as I can tell.

**Claims And Evidence:**

No

**Claims Explanation:**

Please see the comments above. Overall the results are acceptable but I do have a few concerns. The number in effective rank and stable rank does not vary too much. The difference is very subtle. For dormant units, all the mean range is within 0.1%-0.3%. Variation this small really significantly impacts the network performance? I’d like to see a more direct and effective result.

Additionally, the paralysis-ant case, none of the method shows smooth learning behavior. Why?

For the formulation, using MLE instead of Maximum A Posteriori (MAP) in bayesian learning frameworks does not make sense to me. The derivation is a nice touch but the choice of the formulation seems overly complicated and the reasons for this formulation look adjacent to me.

**Requested Changes:**

Please check the comments above. I would like to see
1. More involved discussion of distributional RL, risk sensitive control, and their relationship with the proposed work.
2. Stronger base line that robotics learning people actually use to solve the same problem, e.g., RSL_RL from https://github.com/leggedrobotics/rsl_rl.
3. More direct results showcasing the critic is smoothly changing with changing dynamics.

---

> ### Author Response · Authors · 2025-09-24
> **Response to Reviewer 5uv8 Part 1/3**
>
> We thank the reviewer once again for the valuable feedback. As noted in our general response above, we have made several revisions that strengthen the paper. Below, we provide point-by-point answers to the reviewer's comments.
>
> ## On the adaptation of new tasks
> We have clarified our setting as non-stationary reinforcement learning (see the revised paragraph on “Relation of non-stationary reinforcement learning to meta/continual reinforcement learning.”). In our setting, the underlying control goal remains fixed while only environment dynamics (e.g., friction changes) vary. Therefore, the agent only needs to adjust its policy to current conditions (e.g., modifying torque to cope with higher friction), rather than storing and recalling full policies for all previously encountered dynamics, as it is not expected to encounter past dynamics again. While remembering all scenarios may be beneficial in broader continual learning setups, our work specifically targets rapid, on-the-fly adaptation in perpetually changing environments rather than retaining performance on previously seen dynamics.
>
> ## On the discussion of distributional RL and risk-sensitive control
> - **Relation of (3) to risk-sensitive control:**
> Risk-sensitive RL commonly uses the mean-variance criterion as a risk-averse measure (see, e.g., the discussion in [1] Sections 7.6-7.8), aiming to maximize the expected return while penalizing high-variance returns. Let $G$ be a return random variable, then the mean-variance risk measure $\rho$ is defined as $\rho(G) = \mathbb{E}[G] - \lambda \text{var}[G]$, where $\lambda > 0$.
> UCB [2], as given by (3), instead focuses on an optimistic, epistemic uncertainty-based exploration around, in our case, the generalized advantage estimator $\hat A$.
> Recent work by [3] develops a similar utility function for the critic in an actor-critic setting, which adaptively interpolates between a risk-averse behavior ($\lambda >0$) and a risk-seeking behavior ($\lambda < 0$).
>
> - **Relation to distributional RL:** Our evidential value learning is part of a broader effort to incorporate distributional information into reinforcement learning, addressing two types of uncertainty: aleatoric and epistemic. Traditional distributional RL methods mainly model aleatoric uncertainty for risk-sensitive control, while Bayesian approaches handle epistemic uncertainty for exploration. Evidential value learning differs by using an evidential model over the value function to capture both uncertainties simultaneously, enabling regularization and optimistic exploration. This approach relies on a parameterized density model rather than particle-based approximations, distinguishing it from standard distributional RL techniques.
>
>
> ## On the beginning stages of the experiments.
> While in some cases EPPO variants appear to start with slightly higher performance, there are also counterexamples where PPO matches or even outperforms EPPO at the early stages (e.g., slippery-ant-decreasing). These differences are natural and often arise from the exploration dynamics and adaptability of the models. Overall, across the majority of tasks, PPO and EPPO exhibit comparable performance in the initial stages, with the main differences becoming more pronounced later in training.

---

> > ### Author Response · Authors · 2025-09-24
> > **Response to Reviewer 5uv8 Part 2/3**
> >
> > ## On the metrics, curriculum learning, and results
> > - **Metrics:** Our focus is on sequential tasks where rapid adaptation is crucial. We therefore report both AULC and final return. AULC captures the cumulative reward throughout training and thus reflects adaptation speed and efficiency, while final return measures stable performance after task-specific adaptation. Together, these two metrics highlight complementary aspects of adaptability in non-stationary settings.
> > - **Curriculum learning:** While our setup might resemble curriculum training at first glance, it differs in several major aspects. Task intervals are short and fixed, not determined adaptively by a teacher or completion criterion. The goal is not to gradually progress from easier to harder tasks, but to evaluate how quickly an agent can adapt to arbitrary environmental changes. Moreover, we focus on adaptation to the current task rather than retention of past ones. The task sequences are structured but not designed with a difficulty progression, reflecting changes in environment dynamics rather than a curated curriculum.
> > - **Results:** In the early tasks, all agents perform comparably; the observed differences emerge in how well they adapt as environments change throughout the training. Our experiments use task lengths of 500,000 steps, which is shorter than the 2,000,000 steps used in continual backpropagation [4] or the 1,000,000 steps of the original PPO [5]. With longer training, absolute performance would likely improve, but our interest is specifically in adaptation under constrained time per task. For transparency, we have also provided a notebook implementation of EPPO. We therefore believe the results are not due to implementation issues, but reflect the intended evaluation setup.
> >
> >
> > ## On the dormant unit percentage and plasticity metrics
> > - **Dormant unit percentage:** We thank the reviewer for catching this mistake. In the previous version, we plotted the ratio rather than the percentage. The reported numbers should be multiplied by 100. This correction has been made in the revised version.
> > - **Plasticity metrics:** We use effective rank, stable rank, and dormant unit percentage as standard metrics to assess representational plasticity, following prior work in continual learning. While the absolute differences in these metrics may appear small (especially when aggregated across seeds and experiments in Figure 3), they are consistent and statistically significant across multiple runs and environments, see Figures 5&ndash;7 and Tables 4&ndash;9. Our goal is not to claim that EPPO variants achieve the best possible plasticity, but rather to show that evidential value learning helps mitigate plasticity loss in the critic network compared to plain PPO. Specifically, EPPO_mean improves plasticity relative to plain PPO, and EPPO_cor and EPPO_ind preserve this improvement when combined with directed exploration.
> >
> > ## On the paralysis-ant
> > In these paralysis experiments, we progressively paralyze the joints and later restore their ability to apply torque. At the beginning of the 5th task (2M steps), all joints are fully paralyzed, which naturally causes a sharp performance drop. In subsequent tasks, the joints gradually regain torque ability, leading to the observed drop–recovery pattern. For the ant, which has 8 joints and 4 legs, we paralyze an entire leg (or two legs) at a time, causing severe disruptions in locomotion and thus less smooth learning curves. In contrast, for the cheetah with only 4 joints (2 legs), paralysis is applied at the joint level, so the impact is less drastic. The observed non-smooth behavior is therefore expected and reflects the difficulty of re-coordination under such drastic non-stationarities.

---

> > > ### Author Response · Authors · 2025-09-24
> > > **Response to Reviewer 5uv8 Part 3/3**
> > >
> > > ## On the Maximum Likelihood Estimation and Maximum A Posteriori
> > > We would like to clarify that our approach does not rely on Maximum Likelihood Estimation. Instead, we adopt Type-II Maximum Likelihood (Empirical Bayes) [6], which estimates distributional hyperparameters by maximizing the marginal likelihood of a two-stage data-generating process. This is a well-established alternative to Maximum A Posteriori when priors are parameterized and their values are inferred from data rather than fixed a priori. Our formulation directly follows deep evidential regression [7], where this approach has been shown to provide a practical and stable way to capture both aleatoric and epistemic uncertainty. We therefore believe this choice is natural in our setting, and not an unnecessary complication.
> > >
> > >
> > > ## On the stronger baseline
> > > The most relevant method in RSL_RL is PPO with random network distillation [9], because it studies directed exploration on robotic tasks with changing friction [8]. In our work, we already benchmark against its stronger distributional variant, PPO with DRND [10], which has been shown to outperform PPO with RND [9]. Together with plain PPO, CB, PFO, and EPPO variants, our evaluation covers (i) plasticity preservation (PFO, CB, EPPO_mean) and (ii) directed exploration (PPO_DRND, EPPO_cor, EPPO_ind). Given the breadth of these baselines and the substantial computational cost (~22 GPU-days), we believe our experiments already test the core hypotheses against the strongest state-of-the-art representatives. Nonetheless, we agree that integrating robotics-oriented frameworks such as RSL_RL is an interesting direction for future work and could further broaden the comparison.
> > >
> > >
> > > ## On the changing critic
> > > We refer to our previous response regarding plasticity metrics. We use well-established metrics—effective rank, stable rank, and dormant unit percentage—to measure the plasticity of the critic. Our main claim is not solely about plasticity; rather, we demonstrate that evidential value learning improves the critic’s plasticity over plain PPO. For completeness, we also report comparisons with CB, which is a state-of-the-art method for plasticity preservation but underperforms in adaptation to rapidly changing dynamics. Our key comparisons regarding plasticity, which support our claim, are between plain PPO and EPPO variants. Across all experiments and metrics, EPPO variants show statistically significant improvements ($p < 0.05$), confirming that the critic adapts more effectively to changing dynamics.

---

> > > > ### Author Response · Authors · 2025-09-24
> > > > **References**
> > > >
> > > > [1] Bellemare, et al. *Distributional reinforcement learning*. MIT Press (2023).
> > > >
> > > > [2] Auer, et al. *Finite-time analysis of the multiarmed bandit problem*. Machine Learning (2002).
> > > >
> > > > [3] Tasdighi, et al. *Improving actor-critic training with steerable action-value approximation errors*. ECAI (2025).
> > > >
> > > > [4] Dohare, et al. *Loss of plasticity in deep continual learning*. Nature (2024).
> > > >
> > > > [5] Schulman, et al. *Proximal policy optimization algorithms*. ArXiv (2017).
> > > >
> > > > [6] Efron, et al. *Large-scale inference: Empirical Bayes methods for estimation, testing, and prediction*. Cambridge University Press (2012).
> > > >
> > > > [7] Amini, et al. *Deep evidential regression*. NeurIPS (2020).
> > > >
> > > > [8] Schwarke, et al. *Curiosity-driven learning of joint locomotion and manipulation tasks*. CoRL (2023).
> > > >
> > > > [9] Burda, et al. *Exploration by random network distillation*. ICML (2019).
> > > >
> > > > [10] Yang, et al. *Exploration and anti-exploration with distributional random network distillation*. ICML (2024).

---

> ### Comment · Reviewer_5uv8 · 2025-09-30
>
> I thank the authors for the rebuttal. Overall some of my questions and concerns have been addressed. But there are a few more:
>
> 1. how does your framework address epistemic uncertainty other than using a Bayesian framework? Isn't it true that for distributional RL critic (say quantile regression) can also directly approximate the posterior, instead of relying on specifying a prior and learn the posterior? In that perspective, I don't see significant difference than distributional RL.
>
> 2. The motivation of "fast adaptation" is still unconvincing. it seems the scope of application that truly needs this behavior is restrictive. Besides, to my knowledge, curriculum learning does not need to be difficulty based. People also call episodic change in environment dynamics a curriculum, the same as in this paper.
>
> 3. While the algorithm can be the same, implementation vary vastly from repo to repo. The point I brought up rsl_rl is that, this is a widely used library that can achieve (near) production level performance. Using a custom implementation and treat that as any other PPO implementation does not make sense to me, considering the fact that PPO can be unstable (in my personal experience).

---

> > ### Author Response · Authors · 2025-10-01
> >
> > Thank you for your follow-up questions.
> >
> > ## On the uncertainty and distributional RL
> > >How does your framework address epistemic uncertainty other than using a Bayesian framework?
> >
> > A Bayesian approach to the model in Section 3.1 would consist of inferring a posterior over the latent variables $\omega, \nu, \alpha, \beta, \sigma^2, \mu$ via Monte Carlo methods or variational inference, and then, during prediction, marginalizing over the inferred posterior to obtain a posterior predictive, which could then be used to design a UCB estimator, analogous to ours in Section 3.2.
> > For the scenario we are studying, a full Monte Carlo sampling approach is not computationally tractable, while reverting to a variational approach requires more extensive approximations.
> >
> > Evidential deep learning approaches circumvent these complexities by directly learning second-order epistemic uncertainty through a parametric approach that models distributions over distributions.
> >
> > For a deeper discussion, we refer the reader to the original works in this direction by Sensoy et al. [1] and Malinin & Gales [2], as well as the survey and introduction by Ulmer et al. [3].
> >
> > > Isn't it true that for distributional RL critic (say quantile regression) can also directly approximate the posterior, instead of relying on specifying a prior and learn the posterior?
> >
> > Distributional reinforcement learning [4] does not infer posteriors or epistemic uncertainty. It is concerned with inferring the return distribution, i.e., the aleatoric uncertainty inherent in the MDP.
> >
> > ## On the curriculum learning and fast adaptation
> > We acknowledge that the term curriculum has been used in broader ways, including to describe episodic changes in environment dynamics. However, the central distinction in our work is that our setup is not designed to facilitate learning through a beneficial sequence, but rather to stress-test how quickly an agent can adapt when faced with abrupt, externally imposed changes.
> >
> > In curriculum learning, task order is typically chosen—explicitly or implicitly—to promote transfer or accelerate mastery [5]. In our case, tasks change at short, fixed intervals, independent of success, difficulty, or skill acquisition. The sequences are structured but not pedagogically curated; they reflect arbitrary shifts in environment dynamics rather than a progression intended to guide the learner.
> >
> > The motivation for fast adaptation is that many real-world and research settings—such as robotics in non-stationary environments, agents interacting with humans, or continual learning benchmarks—require agents to adjust rapidly without the luxury of long retraining phases. Our benchmark is therefore not a curriculum in the traditional sense, but a controlled testbed for measuring adaptation speed under dynamic conditions.
> >
> > ## On the implementation
> > We implemented plain PPO ourselves and then built all baselines and our methods on top of this same backbone. This ensures a fair comparison, since any limitation of our PPO implementation equally affects the baselines and our methods. To verify correctness, we compared our PPO performance against external references: for example, at 500k steps in the paralysis-ant environment, our results are comparable to those of CB at the same horizon (see Fig. 3c in [6]), and for paralysis-cheetah, our results are consistent with a highly popular public library's [7] PPO implementation (see [8]). These checks give us confidence that our PPO baseline is representative. Finally, our focus is on fundamental research questions about adaptation to non-stationary environments rather than production-level optimization, so using a lightweight PPO backbone is appropriate for our study.
> > _____
> > **References**
> > [1] Sensoy et al. _Evidential Deep Learning to Quantify Classification Uncertainty_ (2018).
> > [2] Malinin & Gates, _Predictive Uncertainty Estimation via Prior Networks_ (2018).
> > [3] Ulmer et al., _Prior and Posterior Networks: A Survey on Evidential Deep Learning Methods For Uncertainty Estimation_ (2023).
> > [4] Bellemare et al., _Distributional Reinforcement Learning_ (2023).
> > [5] Bengio, et al. *Curriculum learning*. ICML (2009).
> > [6] Dohare, et al. *Loss of plasticity in deep continual learning*. Nature (2024).
> > [7] Huang, et al. *Cleanrl: High-quality single-file implementations of deep reinforcement learning algorithms*. JMLR (2022).
> > [8] https://wandb.ai/costa-huang/cleanRL/reports/MuJoCo-v4-CleanRL-s-PPO--VmlldzozMTIxOTI5

---

> > > ### Comment · Reviewer_5uv8 · 2025-10-01
> > >
> > > Again, I thank the authors for the detailed response. I think the explanation makes the story much clearer and now all makes sense to me. I encourage the authors to include the discussions into the paper, which will make your arguments and claims more convincing. I will suggest acceptance.

---

> > > > ### Author Response · Authors · 2025-10-02
> > > >
> > > > We thank the reviewer for the constructive feedback and positive recommendation.

---

### Author Response · Authors · 2025-09-24
**General Response to Reviewers**

We thank the reviewers for their constructive and insightful feedback, which helped us improve the quality and clarity of the paper. Below, we summarize the main changes made in the revised version.

## Summary of Revisions

- **New Baseline --- Continual Backpropagation (CB) [1]:** Added a continual learning baseline to strengthen the empirical comparison.  While CB maintains high plasticity, it fails to adapt effectively to non-stationary environments.
- **Non-stationary vs Continual Learning:** Reframed the problem and clarified the differences between the two settings.
- **Extended Related Work:** Expanded the discussion on directed exploration, loss of plasticity, and continual learning.
- **Corrections:** Corrected the dormant unit percentage figures and fixed citation errors.
- **Significance Test:** Added statistical significance tests to the results table and provided further details in the appendix.
- **Hyperparameter Tuning:** Justified hyperparameter choices for the baselines and conducted a grid search for the new baseline.
- **Evidential Deep Learning Details:** Provided additional explanation and details in the evidential deep learning section.


We believe these changes significantly strengthen the manuscript and address the reviewers’ concerns. Detailed point-by-point responses are provided in the following sections. In case it is helpful, we did not mark changes in the PDF, as OpenReview provides a _compare revisions_ feature that highlights the differences between versions.


**References**

[1] Dohare, et al. *Loss of plasticity in deep continual learning*. Nature (2024).

---

### Author Response · Authors · 2025-10-17
**General Response to Reviewers**

We thank the reviewers for their engagement during the discussion period, which helped improve the quality and clarity of the paper. Below, we summarize the main updates made in the current revised version compared to the previous version.

- We clarified the relationship between our non-stationary RL setup, curriculum learning, and continual reinforcement learning in Section 2.1.
- We added an empirical analysis of the exploration capabilities of PPO, PPO_DRND, and all EPPO variants (Table 5, Section 4.1, Appendix B.1.4).
- We expanded the discussion of plasticity preservation through evidential value learning, including its impact on gradient scaling and loss-surface smoothing (Section 4.1).
- We provided additional diagnostics tracking effective rank, stable rank, dormant units, and critic max gradient norms throughout training (Figures 8 and 9).

We believe these changes significantly strengthen the manuscript. For convenience, we did not mark changes in the PDF, as OpenReview provides a _compare revisions_ feature that highlights differences between versions.

___
Update (22.10.2025): We extend the _Loss of plasticity mitigation_ paragraph in the _Background_ section.

---

### Decision · Action_Editor_2U2C · 2025-10-27

**Recommendation:** Accept as is

**Audience:**

Yes

**Audience Explanation:**

This work is of interest to the general reinforcement learning community.

**Claims And Evidence:**

Yes

**Claims Explanation:**

The authors provide adequate empirical evidence to support the claims; this is particularly true after addressing the feedback from the reviewers, where more experiments and discussion were added to strengthen the evidence of the claims.